# Non-coding autoimmune risk variant defines role for ICOS in T peripheral helper cell development

Taehyeung Kim[1], Marta Martínez-Bonet[2,3], Qiang Wang [1], Nicolaj Hackert [1,4,5,6], Jeffrey A. Sparks [2], Yuriy Baglaenko[2,4], Byunghee Koh[2], Roxane Darbousset [1], Raquel Laza-Briviesca [1], Xiaoting Chen[7], Vitor R. C. Aguiar [1,4], Darren J. Chiu[2], Harm-Jan Westra [2,4,8], Maria Gutierrez-Arcelus[1,4], Matthew T. Weirauch [7,9,10], Soumya Raychaudhuri[2,4], Deepak A. Rao [2] & Peter A. Nigrovic [1,2] ✉

Fine-mapping and functional studies implicate rs117701653, a non-coding single nucleotide polymorphism in the *CD28/CTLA4/ICOS* locus, as a risk variant for rheumatoid arthritis and type 1 diabetes. Here, using DNA pulldown, mass spectrometry, genome editing and eQTL analysis, we establish that the disease-associated risk allele is functional, reducing affinity for the inhibitory chromosomal regulator SMCHD1 to enhance expression of inducible T-cell costimulator (ICOS) in memory CD4+ T cells from healthy donors. Higher ICOS expression is paralleled by an increase in circulating T peripheral helper (Tph) cells and, in rheumatoid arthritis patients, of blood and joint fluid Tph cells as well as circulating plasmablasts. Correspondingly, ICOS ligation and carriage of the rs117701653 risk allele accelerate T cell differentiation into CXCR5-PD-1high Tph cells producing IL-21 and CXCL13. Thus, mechanistic dissection of a functional non-coding variant in human autoimmunity discloses a previously undefined pathway through which ICOS regulates Tph development and abundance.

Rheumatoid arthritis (RA) and type 1 diabetes (T1D) are prevalent autoimmune diseases in which immune attack leads to permanent tissue injury[1,2]. Both diseases are commonly accompanied by autoantibodies, including rheumatoid factor and anti-citrullinated protein antibodies (ACPA) in RA[3] and anti-islet cell, anti-glutamic acid decarboxylase, and anti-insulin antibodies in T1D[4]. B cells producing autoantibodies are readily identified in rheumatoid joints[5]. B cell-T cell interactions are also implicated in T1D[6,7]. These findings highlight the importance of mechanisms that underlie disordered immune tolerance in human autoimmunity.

Both RA and T1D are highly polygenic. Genome-wide association studies (GWAS) have identified 124 risk loci for RA and at least 53 for T1D[8,9]. Many of these loci are relevant to CD4+ T cell function, suggesting a key role for aberrant T cell help[10–12]. Unfortunately, few GWAS

[1]Division of Immunology, Boston Children's Hospital, Harvard Medical School, Boston, MA, USA. [2]Division of Rheumatology, Inflammation, and Immunity, Brigham and Women's Hospital, Harvard Medical School, Boston, MA, USA. [3]Laboratory of Immune-regulation, Instituto de Investigación Sanitaria Gregorio Marañón, Madrid, Spain. [4]Broad Institute of MIT and Harvard, Cambridge, MA, USA. [5]Division of Rheumatology, Department of Medicine V, Heidelberg University Hospital, Heidelberg, Germany. [6]Institute for Immunology, Heidelberg University Hospital, Heidelberg, Germany. [7]Center for Autoimmune Genomics and Etiology, Cincinnati Children's Medical Center, Cincinnati, OH, USA. [8]Department of Genetics, University Medical Center Groningen, University of Groningen, Hanzeplein 1, Groningen, The Netherlands. [9]Divisions of Human Genetics, Biomedical Informatics, and Developmental Biology, Cincinnati Children's Hospital Medical Center, Cincinnati, OH, USA. [10]Department of Pediatrics, University of Cincinnati College of Medicine, Cincinnati, OH, USA. ✉e-mail: peter.nigrovic@childrens.harvard.edu

hits have been solved definitively, even to the level of the affected gene. This deficit, especially remarkable given the decades since GWAS methodology was first introduced, reflects the difficulty of linking noncoding variants to function. Dissecting such variants is bioinformatically and experimentally challenging because functional noncoding single nucleotide polymorphisms (SNPs) represent a very small fraction of all disease-associated SNPs, have small effect sizes, and often exhibit their activity only in specific lineages and/or activation states[13,14].

We had previously employed Bayesian fine mapping to prioritize a set of non-coding variants shared by RA and T1D as likely functional[15]. These included rs117701653, a noncoding biallelic SNP in the *CD28/CTLA4/ICOS* region on chromosome 2. CD28 is a costimulatory receptor that provides a key second signal to T cells activated via the T cell receptor[16]. CTLA4 attenuates co-stimulation by competing with CD28 for CD80 and CD86 and by stripping these molecules from the surface of antigen-presenting cells[17,18]. ICOS (inducible T cell co-stimulator) is expressed by activated T cells and recognizes a distinct counter-receptor, termed ICOS ligand, to modulate Th1, Th2, and Th17 responses and promote T-dependent antibody formation[19,20]. Despite the proximity of rs117701653 to these critical T cell genes, how this SNP modulates the risk of systemic autoimmunity remains unknown.

Here, we demonstrate that rs117701653 modulates *ICOS* expression through allelic affinity for the inhibitory chromatin regulator SMCHD1 (structural maintenance of chromosomes flexible hinge domain-containing protein 1). The risk allele A reduces binding to SMCHD1 compared with the protective allele C, leading to greater expression of ICOS by CD4$^+$ memory T cells. In turn, ICOS accelerates the development of T peripheral helper (Tph) cells, a specialized population of B cell-helper CD4$^+$ T cells implicated in RA and T1D[21,22], in a manner reflected in primary T cells from healthy donors carrying the risk allele at rs117701653. Thus, a genetic risk variant for human autoimmunity exposes a previously unrecognized pathway regulating the development and abundance of a pathogenic T helper cell population.

## Table 1 | Identification of proteins binding to rs117701653 SNP by FREP

| Identified protein | Molecular weight | Peptide count | | |
|---|---|---|---|---|
| | | Bio-rs1177 | Bio-rs1177 + Competitor | Bio-Ctrl |
| ALB | 69 kDa | 14 | 26 | 23 |
| PARP1 | 113 kDa | 14 | 17 | 26 |
| DST | 861 kDa | 3 | 1 | 6 |
| RYR1 | 565 kDa | 3 | 1 | 2 |
| **H3BNH8** | **10 kDa** | **3** | **0** | **0** |
| ACACB | 277 kDa | 3 | 0 | 1 |
| **SMCHD1** | **226 kDa** | **3** | **0** | **0** |
| SYNE1 | 1011 kDa | 2 | 0 | 5 |
| DYNC1H1 | 532 kDa | 2 | 1 | 3 |
| GOLGB1 | 376 kDa | 2 | 0 | 4 |
| SF3B1 | 146 kDa | 2 | 0 | 3 |
| TOP1 | 91 kDa | 1 | 3 | 1 |
| SMG1 | 410 kDa | 1 | 0 | 4 |
| ROCK1 | 158 kDa | 1 | 0 | 3 |
| SPEN | 402 kDa | 1 | 0 | 3 |

Flanking restriction enhanced pulldown (FREP) was performed using biotinylated (Bio-rs1177) or non-biotinylated (Competitor) DNA fragment containing C-allele of rs117701653 SNP. Biotinylated DNA fragment of irrelevant sequence (Bio-Ctrl) was used as a negative control. Peptides were analyzed by mass spectrometry, identifying 43 proteins (Supplementary Table 1). 15 proteins with more than one peptide fragment in Bio-rs1177 are listed. Bolded H3BNH8 and SMCHD1 indicate selective binding to rs117701653.

## Results

### rs117701653 allelically modulates binding of the chromatin regulator SMCHD1

Prior studies of rs117701653 found greater binding of Jurkat T cell nuclear extract protein to the protective C allele (frequency in EUR = 0.06) than to the risk A allele (allele frequency in EUR = 0.94)[15]. To identify this protein or protein complex, we applied an efficient DNA pulldown technique, flanking restriction enhanced pulldown (FREP)[23], using nuclear extract from Jurkat T cells and bait DNA corresponding to the C allele of rs117701653. Mass spectrometry identified 43 candidate proteins (Table 1, Supplementary Table 1). Of these, 41 displayed binding despite the presence of a competitor and/or nonspecific binding to negative control, leaving two that bound selectively to rs117701653: the uncharacterized protein H3BNH8 and the chromatin regulator SMCHD1. Since SMCHD1 is known to modulate gene expression via interaction with chromatin, we selected this protein for further analysis[24,25].

To confirm that SMCHD1 binds rs117701653, we performed western blot on the nuclear proteins released from FREP. Anti-SMCHD1 recognized a band of the appropriate size, approximately 225 kDa, competed away by a non-biotinylated rs117701653 oligonucleotide (Fig. 1A). Further, by electrophoretic mobility shift assay (EMSA), three different anti-SMCHD1 antibodies inhibited binding to the C allele oligonucleotide when applied to Jurkat nuclear extract before addition of biotinylated probe, or alternately induced a supershift when applied to pre-mixed nuclear extract and probe (Fig. 1B, Supplementary Fig. 1). Consistent with this observation, ChIP-qPCR in human peripheral blood mononuclear cells (PBMC) using anti-SMCHD1 enriched strongly for rs117701653 ($P = 0.014$), an effect comparable in magnitude to a SMCHD1-binding positive control, the HS17 promoter (Fig. 1C). Finally, we applied CRISPR-mediated homology-directed repair (HDR) in Jurkat T-cells to convert the wild-type allele A to the modified allele C, generating seven clones homozygous for the wild-type A allele, three heterozygous clones, and three clones homozygous for the modified allele C. ChIP-qPCR confirmed increased SMCHD1 binding to rs117701653 in the C/C modified-allele clones compared to the A/A clones ($P = 0.0083$; Fig. 1D). Together, these observations establish that rs117701653 binds SMCHD1 and that the C allele enhances binding relative to the A allele.

### SMCHD1 binding at rs117701653 regulates expression of ICOS

Next, we sought to establish whether SMCHD1 modulates transcription of a gene in the *CD28/CTLA4/ICOS* locus (Fig. 2A). We therefore performed low-input RNA-seq in resting total CD4$^+$ T cells from 24 healthy subjects with 8 A/A, 8 A/C, and 8 C/C genotypes at rs117701653 and analyzed association with expression of all 11 protein-coding genes lying within 1 Mb of the SNP. Surprisingly, expression of *CD28* and *CTLA4* did not vary with genotype, nor did *RAPH1*, another nearby gene[26]. Instead, allelic variation at rs117701653 correlated strongly with expression of *ICOS*, located 173 kb 3′ (β = −0.74 and $p = 0.0019$ by a linear model corrected for age, sex, and rank-normal transformed residuals) (Fig. 2A, B, Supplementary Fig. 2). Correspondingly, Jurkat clones modified to carry the C allele exhibited lower *ICOS* transcript and ICOS protein than those bearing the wild-type A/A genotype (Fig. 2C; Jurkat cells do not express *CTLA4*). Whereas CD28 unexpectedly displayed a similar trend, we tested related downstream signaling pathways[27,28], and found that A/A and C/C clones exhibited no significant difference in signaling downstream of anti-CD3/CD28 stimulation but that C/C clones stimulated with anti-CD3/ICOS displayed lower AKT phosphorylation, a pathway engaged by ICOS (Supplementary Fig. 3).

Since SMCHD1 typically represses gene expression[24,25], we hypothesized that enhanced binding to the rs117701653 C allele would suppress *ICOS*. The genomic context of rs117701653 rendered the efficiency of CRIPSR-mediated HDR too low (0.06% for generation of

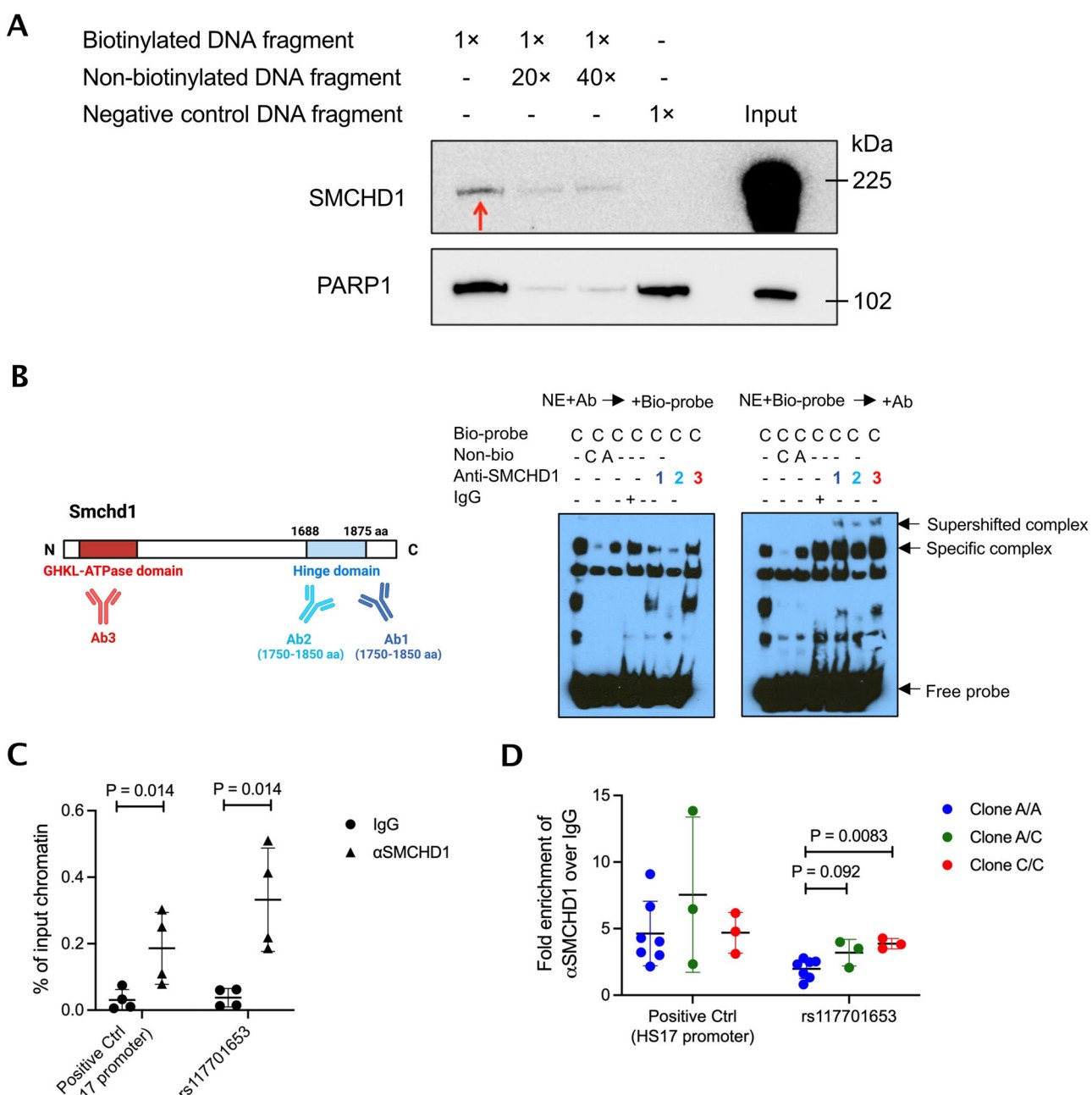

**Fig. 1 | Demonstration of the allele-specific binding of SMCHD1 to SNP rs117701653. A** Validation of SMCHD1 binding to rs117701653 C allele. Western blot was performed with proteins eluted from FREP and antibodies against SMCHD1 and PARP1 using biotinylated DNA fragment containing rs117701653 C allele as bait. Non-biotinylated DNA fragment and biotinylated DNA fragment of irrelevant sequence served as competitor and negative control respectively. The red arrow indicates specific binding of SMCHD1. PARP1 identified by mass spectrometry showed non-specific binding to negative control DNA fragment. The western blot is representative of three independent biological replicates. **B** EMSA with Jurkat nuclear extract using DNA probe containing rs117701653 C allele and three anti-SMCHD1 antibodies. A diagram illustrates C-terminal hinge domain of SMCHD1 recognized by antibodies 1 and 2 (purple and blue) and N-terminal ATPase domain recognized by antibody 3 (red) (created with BioRender.com). The antibodies were incubated with the nuclear extract before forming protein-DNA probe complex (left gel) or added after the formation (right gel). The EMSA blot is representative of three independent biological replicates. A flow diagram in Supplementary Fig. 1 depicts the method. **C** PBMCs from 4 donors and (**D**) CRISPR-edited Jurkat clones were used for ChIP-qPCR with primer sets targeting rs117701653 SNP and positive control HS17 promoter region[34]. A total of 13 biologically independent clones were generated by CRISPR-Cas9 and ssDNA oligonucleotide HDR template: 7 clone homozygous for wild-type A allele (blue), 3 heterozygous (green), and 3 homozygous for modified C allele (red). Mean ± S.D, *P* value from Mann-Whitney one-tailed *U*-test. Uncropped western blots are included in the Source Data file.

A/C or C/C clones) to apply to primary T cells[29]. Therefore, we generated SMCHD1 knockout clones from our HDR-edited A/A and C/C clones, finding increased ICOS protein only in the C/C context, confirming that SMCHD1 binding to the rs117701653 C allele represses ICOS expression (*P* = 0.031) (Fig. 2D).

## rs117701653 correlates with abundance of circulating Tph cells

Disease-associated genetic variants regulate not only protein expression but also immune cell abundance[30,31]. SMCHD1 is expressed in a variety of immune cell types, including CD4+ and CD8+ T cells, B cells, monocytes, and NK cells. ICOS is expressed mainly in subsets of CD4+

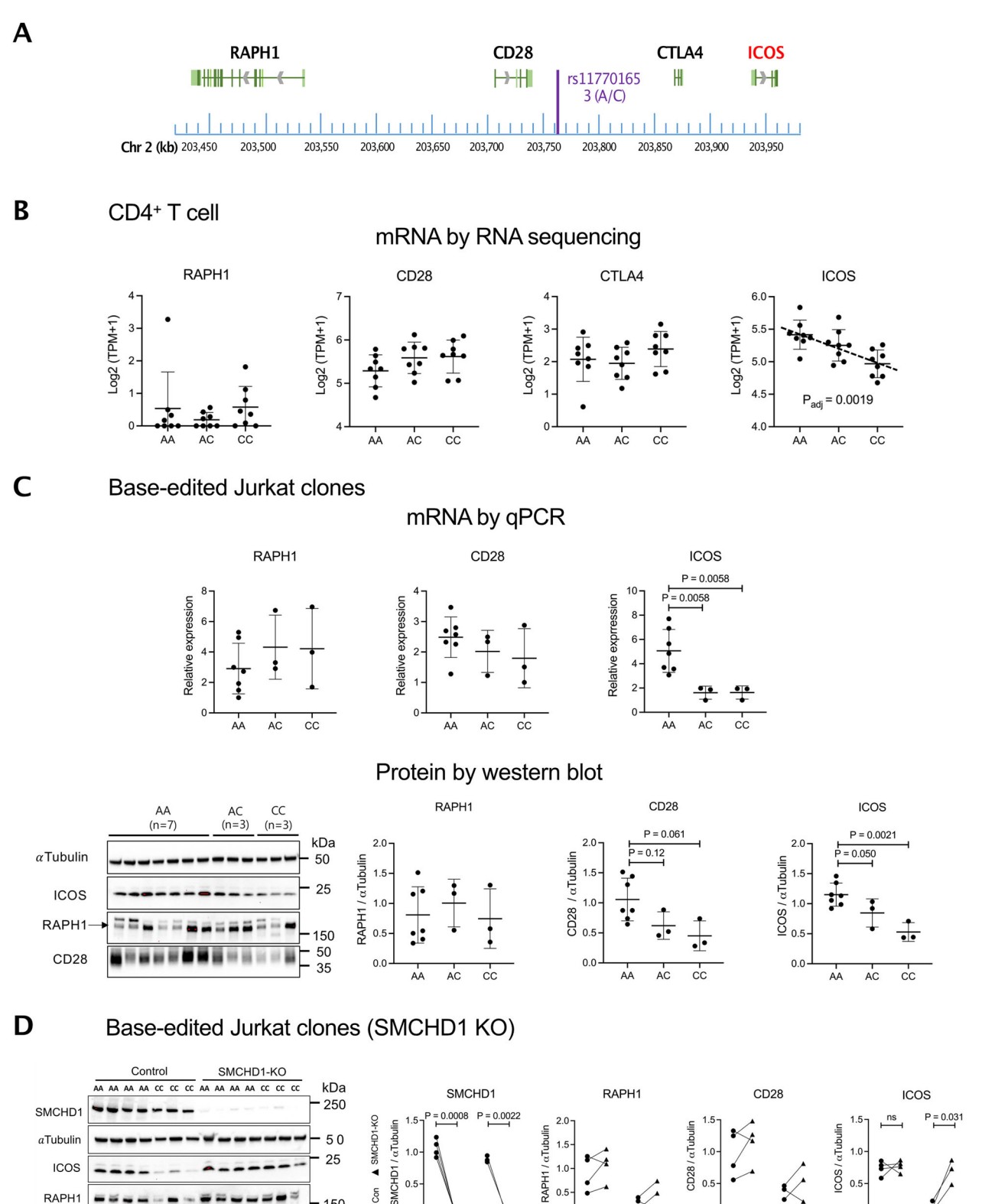

T cells (Supplementary Fig. 4). We hypothesized therefore that a rs117701653-SMCHD1-ICOS pathway would likely be most evident in CD4$^+$ T cells, and so evaluated changes in CD4$^+$ T cell subset abundance as a function of genotype at rs117701653[10,11,32]. Circulating Treg, Th1, Th2, Th17, Tfh, and Tph cells were enumerated in 46 genotyped healthy adults by flow cytometry (Supplementary Fig. 5). Across all subjects, the proportion of cells that expressed ICOS was higher

in CD3$^+$CD4$^+$CD45RA$^-$ memory CD4$^+$ T cells (0.5-3.9%) than in CD3$^+$CD4$^+$CD45RA$^+$ naïve CD4$^+$ T cells, where it was barely detectable (0.0−0.4%) (Fig. 3A, B). As per our earlier findings, the proportion of ICOS$^+$ memory CD4$^+$ T cells was higher in subjects homologous for the A allele at rs117701653 (memory T cells, β = −0.42, $P$ = 0.0097 by linear regression with adjustment for age and sex) (Fig. 3A, B, Supplementary Fig. 6). No genotype-associated difference in ICOS$^+$

**Fig. 2 | Characterization of functional rs117701653 SNP by CRIPSR-mediated editing in Jurkat T cells. A** Map of SNP rs117701653 at *RAPH1-CD28-CTLA4-ICOS* locus on human chromosome 2q33: chromosome position (blue), gene transcripts (black/red), SNP rs117701653 (purple). **B** mRNA levels by low-input RNA sequencing in resting CD4[+] T cells from 24 healthy donors ($n = 8$ A/A, $n = 8$ A/C, and $n = 8$ C/C genotypes at rs117701653), **C** Biologically independent HDR-edited Jurkat clones ($n = 7$ A/A, $n = 3$ A/C, and $n = 3$ C/C), and (**D**) SMCHD1-deleted clones from biologically independent wild-type or edited clones ($n = 4$ A/A and $n = 3$ C/C). CTLA-4 was nondetectable in Jurkat cell clones. Western blot detected proteins expression of $\alpha$-Tubulin (55 kDa), SMCHD1 (225 kDa), ICOS (22 kDa), RAPH1 (195 kDa), CD28 (40-60 kDa). Error bars represent mean ± S.D. For cis-eQTL mapping with the resting CD4[+] T cells, we targeted 11 protein-coding genes with transcription start sites within a 1 Mb window of rs117701653 (*ICA1L, WDR12, CARF, NBEAL1, CYP20A1, ABI2, RAPH1, CD28, CTLA4, ICOS,* and *PARD3B*) (Supplementary Fig. 2). *P* values were computed using a linear model by QTLtools for association between genotypes and expression levels corrected for multiple comparison, age, sex, and rank-normal transformed residuals (**B**). *P* values from one-way ANOVA corrected for multiple comparison by FDR using two-stage linear step-up procedure of Benjamini, Krieger and Yekutieli (**C**), or paired t-test with two-tailed significance (**D**). Uncropped western blots are included in the Source Data file.

proportion was noted in Tph, Tfh, Th1, Th2, Th17, or Treg cells (Supplementary Fig. 7).

Intriguingly, allelic variation at rs117701653 correlated with the proportion of CXCR5[-]PD-1[high] Tph cells, with A/A subjects showing more Tph cells than A/C and C/C subjects ($\beta = -0.60$, $P = 0.010$ by linear regression with adjustment for age and sex) (Fig. 3C, D). Further, the proportion of memory T cells expressing ICOS correlated directly with the proportion of Tph cells (Pearson $\rho = 0.44$ (95% CI, 0.17 to 0.65), $P = 0.0023$; Fig. 3E), a relationship not achieving statistical significance in other subsets though trending similarly in Tfh and Treg cells (Pearson correlation; Tfh $\rho = 0.29$ ($-0.0025$ to 0.53), Th1 $\rho = -0.032$ ($-0.32$ to 0.26), Th2 $\rho = 0.00067$ ($-0.29$ to 0.29), Th17 $\rho = 0.10$ ($-0.1$ to 0.38), Treg $\rho = 0.28$ ($-0.014$ to 0.53)) (Fig. 3E, Supplementary Fig. 8A). Since pathways of Tph development remain poorly understood, these observations prompted us to consider whether the rs117701653-SMCHD1-ICOS axis modulates Tph abundance.

## ICOS expression in memory T cells correlates with Tph and plasmablast B-cell frequency in RA

Tph cells are expanded in RA, helping to sustain pathogenic B cells in the inflamed joint[21]. We sought to confirm the relationship between ICOS expression in memory CD4[+] T cells and Tph abundance. To this end, we obtained mass cytometry data for circulating PBMC from 27 RA patients and 18 controls from the Accelerating Medicines Partnership in RA and Lupus[33]. Both RA and control samples exhibited comparable proportions of CD4[+] T cells that were CD45RA[-]CD45RO[+] memory T cells and of these cells that expressed ICOS (Fig. 4A,B). As observed in our healthy donors, the proportion of ICOS[+] CD4[+] memory cells varied directly with the proportion of CD4[+] T cells that were CXCR5[-]PD-1[high] Tph, an effect evident in both RA patients (Pearson $\rho = 0.64$, $P = 0.00040$) and controls (Pearson $\rho = 0.60$, $P = 0.0089$) (Fig. 4C, Supplementary Fig. 8B, Supplementary Fig. 9). For CXCR5[+]PD-1[high] Tfh cells (Fig. 4D), a correlation was noted in both RA patients (Pearson $\rho = 0.57$, $P = 0.0021$) and controls ($\rho = 0.55$, $P = 0.020$). A previous report analyzing these data observed a correlation between the proportion of memory CD4[+] T cells that were Tph cells and the proportion of plasmablasts among B cells[34]. Correspondingly, in RA patients, the proportion of memory CD4[+] cells expressing ICOS correlated directly with the proportion of CD19[+] B cells that were CD45[+]CD19[+]CD20[-]CD38[high]CD27[+] plasmablasts (Pearson $\rho = 0.58$, $P = 0.0026$) (Fig. 4E).

We then examined cells in synovial fluid from RA joints, where Tph cells are more abundant than in peripheral blood[21]. As expected, compared with healthy donor blood, a 1.7-fold higher proportion of synovial fluid CD4[+] T cells were CD3[+]CD4[+]CD45RA[-] memory cells, and ~20% of memory cells were Tph (Fig. 4F, G, Supplementary Fig. 5 for gating strategies and other memory subsets). RA synovial fluid displayed a clear relationship between the proportion of ICOS[+] memory T cells and Tph abundance, not evident for other subsets tested, including Tfh cells (Fig. 4H; Supplementary Fig. 8C shows that this effect was preserved as a trend when considering only non-Tph CD4[+] memory T cells). Together, these findings

further confirm the association between ICOS and the Tph cell population.

## ICOS stimulation complements TGF-$\beta$ to induce Tph differentiation

Tph cells are characterized by a CXCR5[-]PD-1[high] surface phenotype and express the B cell chemoattractant CXCL13 and the plasmablast differentiation factor IL-21[21]. To test whether ICOS accelerates Tph development, we isolated healthy donor memory CD4[+] T cells via negative selection and induced further differentiation with anti-CD3/CD28 beads together with TGF-$\beta$, ICOS stimulation via anti-ICOS, or both (Fig. 5A). TGF-$\beta$ induced expression of PD-1 (day 6, 10, 14, 18), as expected[35,36] (Fig. 5B, Supplementary Fig. 10A). In isolation, ICOS ligation failed to induce either PD-1 or CXCR5; however, when combined with TGF-$\beta$, ICOS ligation accelerated PD-1 expression, enhancing the generation of CXCR5[-]PD-1[high] cells early (day 3) and again late (day 18), without comparable impact on the development of CXCR5[+]PD-1[high] Tfh-like cells (Fig. 5C, D, Supplementary Fig. 10B, C). Compared with TGF-$\beta$ alone, ICOS ligation amplified IL-21 production early and CXCL13 production late (Fig. 5E, F, Supplementary Fig. 10D–F). Thus, as suggested by our observational findings, ICOS ligation promotes the development of the Tph phenotype.

## The rs117701653 risk allele accelerates Tph development

To test the impact of rs117701653, we differentiated Tph from memory CD4[+] T cells isolated from 46 genotyped healthy subjects. After 3 days of culture with TGF-$\beta$ and ICOS ligation, the percentage of cells expressing this marker was highest in A/A cells (ICOS % $\beta = -2.2$, $P = 0.0028$ by linear regression with age and sex) (Fig. 6A). A/A donors generated more CXCR5[-]PD-1[high] cells (CXCR5[-]PD-1[high] cell % $\beta = -4.1$, $P = 0.0020$ by linear regression with adjustment for age and sex), whereas no consistent effect was seen for CXCR5[+]PD-1[high] cells (Fig. 6B). IL-21 expression, peaking under these culture conditions at day 3 as noted above, was also greatest in A/A cells (IL-21 MFI $\beta = -43.5$, $P = 0.0041$ by linear regression with adjustment for age and sex), and exhibited a clear correlation (Pearson $\rho = 0.39$, $P = 0.0096$) with ICOS MFI across all subjects (Fig. 6C). No comparable effect was observed beginning from naïve CD4 + T cells (Supplementary Fig. 11). Of note, after prolonged stimulation for 10 and 18 days, ICOS no longer exhibited a genotype-driven difference, likely reflecting saturation, although the relationship between ICOS expression and the late cytokine CXCL13 remained (Fig. 6D–F, Supplementary Fig. 12). Similarly, in memory CD4 + T cells, SMCHD1 deletion elevated expression of ICOS and IL-21, but not CD28, in C/C but not A/A cells, confirming the allele-dependent SMCHD1/ICOS relationship in primary cells (Fig. 6G–H, Supplementary Fig. 13). These findings confirm that a genetic variant that enhances ICOS expression correspondingly facilitates T cell differentiation into potentially pathogenic Tph cells.

## Discussion

GWAS have identified an abundance of loci associated with polygenic human phenotypes but translation of these hits into mechanistic insight remains limited. Linkage disequilibrium complicates efforts to

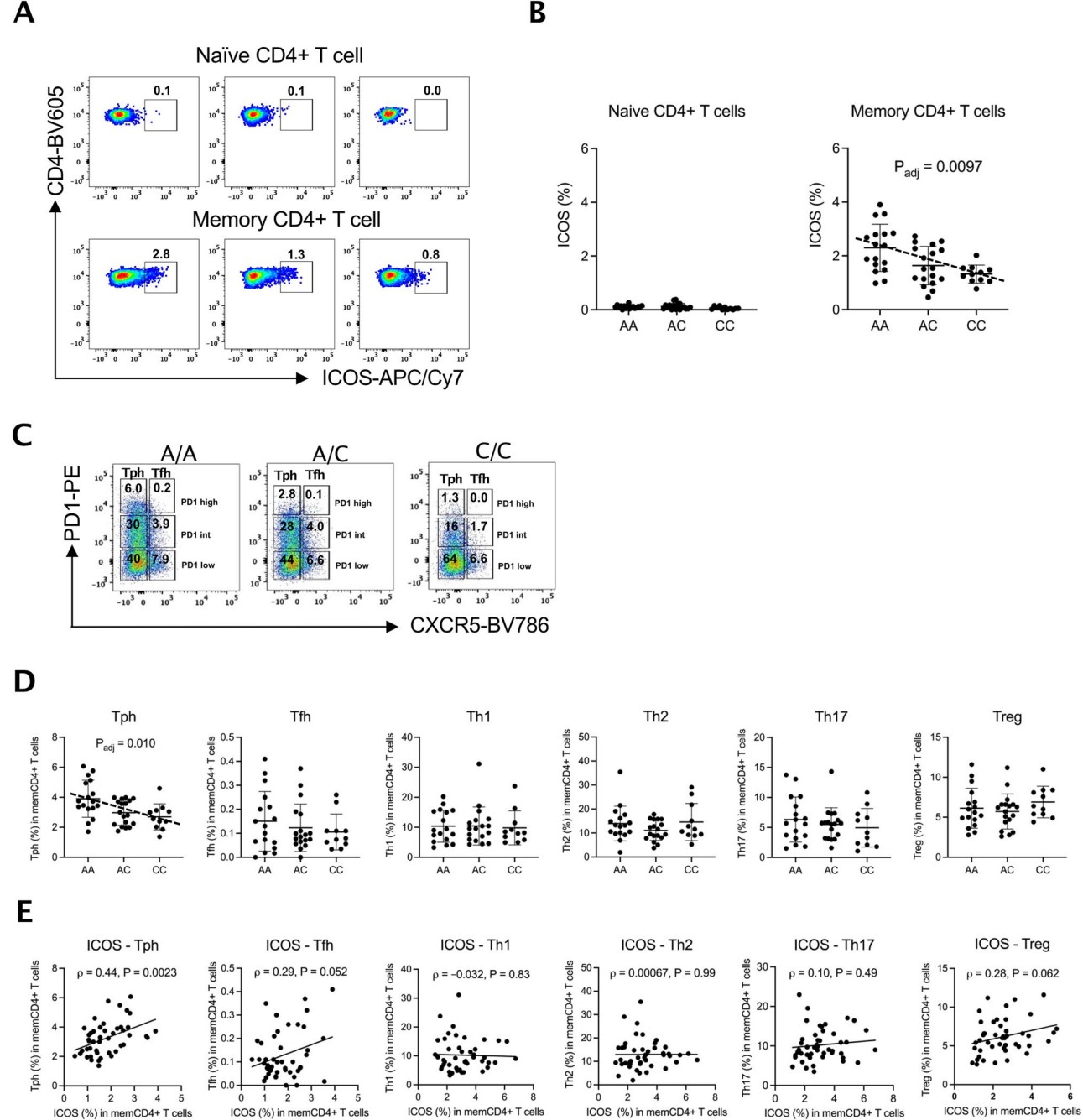

**Fig. 3 | SNP rs117701653 correlates with ICOS expression and circulating Tph cell frequency.** The frequency of ICOS expression among CD4⁺ T cells subsets was determined by flow cytometry for 46 healthy subjects with A/A (n = 17), A/C (n = 18), and C/C (n = 11) genotype at SNP rs117701653. **A** ICOS expression frequency in naïve and memory CD4⁺ T cells. **B** Frequency of ICOS⁺ cells by genotype in naïve and memory CD4⁺ T cells for all 46 donors. **C** Example of gating applied to determine CXCR5⁻PD1^high as Tph and CXCR5⁺PD1^high as Tfh from A/A, A/C, and C/C subjects. **D** Frequency of memory CD4⁺ T cells subsets by genotype at rs117701653 for all 46 donors. **E** Correlation between frequency of ICOS⁺ memory CD4 + T cells and frequency of memory CD4⁺ T cells subsets across all 46 donors. Error bars are Mean ± S.D. *P* values determined using a linear regression model adjusted for age and sex (**B**, **D**). Pearson correlation with a two-tailed test (**E**).

use statistical methods to distinguish the disease-associated functional variant from co-segregating incidental variants. Non-coding variants do not always reside adjacent to their target genes, hindering efforts to elucidate which genes are actually regulated, in particular since expression quantitative trait loci (eQTL) studies remain limited with respect to the lineages and activation states for which data are available[13]. Even where a functional non-coding variant and its gene target have been identified, related regulatory mechanisms and downstream traits often remain obscure, especially for complex diseases such as RA and T1D in which multiple cell types play key

pathogenic roles. Here we overcome these hurdles, confirming rs117701653 as a functional non-coding variant in the immunologically critical *CD28/CTLA4/ICOS* locus and demonstrating its role in a previously unrecognized pathway governing Tph cell development, bridging the gap between GWAS and mechanism.

The sequence of investigations required to establish this conclusion highlights the complexity of identifying, confirming, and understanding common non-coding variants. rs117701653 was nominated through Bayesian fine-mapping of 11,475 RA cases, 9,334 T1D cases, and 26,981 controls, together with preliminary experimental

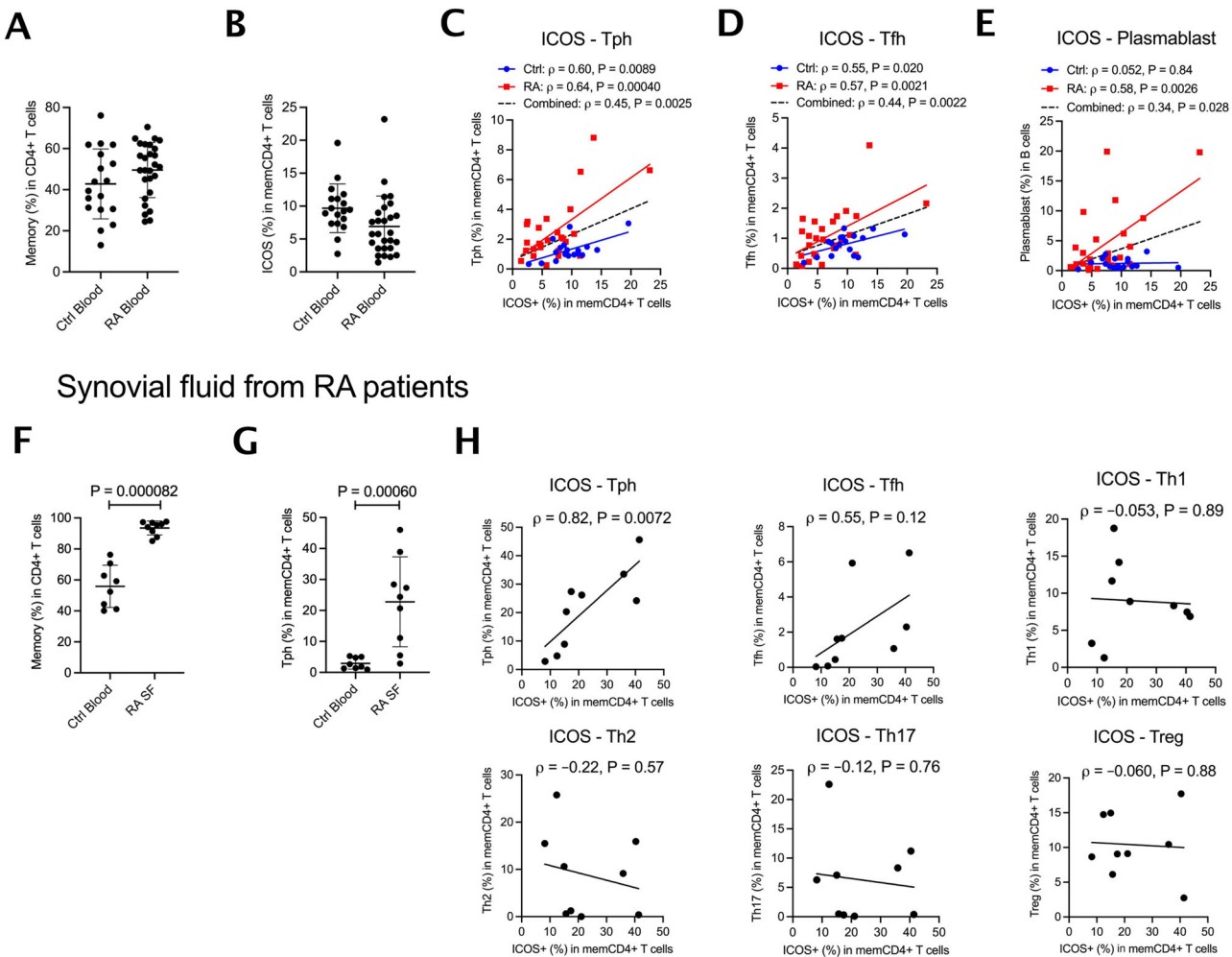

**Fig. 4 | ICOS expression in memory T cells correlates with Tph cell and plasmablast B-cell frequency in rheumatoid arthritis patients.** Mass cytometry data were analyzed from PBMCs of 18 healthy control and 27 RA patients. **A** Frequency of memory cells in CD4[+] T cells and (**B**) ICOS+ cells among memory CD4 + T cells in 45 donors. **C**–**E** Correlation between frequency of ICOS[+] cells among memory T cells and frequency of Tph among memory T cells, Tfh among memory T cells, or plasmablasts among B cells. Flow cytometry using PBMCs from blood of 8 healthy controls (blue) and synovial fluid of 9 RA patients (red) were examined for ICOS expression and Tph cells proportion. **F** Frequency of memory cells in CD4[+] T cells and **G** Tph cells in memory T cells in 17 donors. **H** From synovial fluid of RA patients, correlation between the frequency of ICOS among memory T cells and the frequency of subsets including Tph cells among memory T cells. Error bars are mean ± S.D. *P* values from Mann-Whitney two-tailed *U*-test (**F, G**). Pearson correlation with a two-tailed test (**C–E, H**).

confirmation by EMSA and luciferase expression[15]. Here we confirmed this result through definition of the responsible regulatory protein, SMCHD1, using FREP, mass spectrometry, supershift, chromatin immunoprecipitation, and HDR-editing at rs117701653. *ICOS* was established as the target gene through RNA sequencing of healthy donors recruited based on genotype at rs117701653, with the role of this SNP in the regulation of *ICOS* by SMCHD1 further confirmed through HDR editing and CRISPR-based SMCHD1 deletion in Jurkat clones and primary T cells. Observing that carriage of the A risk allele corelated with higher ICOS expression in CD4[+] memory T cells in healthy donors, and that ICOS expression paralleled Tph abundance in both and RA patients as well as plasmablast frequency in RA, we show that ICOS ligation accelerates Tph cell development, replicated as a function of rs117701653 in healthy donors.

Considered in isolation, allelic variation at rs117701653 plays a minor role in population disease burden, since the protective C allele frequency is only 6% in the EUR population and carries an odds ratio of 0.74 for RA and 0.79 for type 1 diabetes[15]. However, like rare variants that alter protein sequence, functional noncoding variants are important because they are natural experiments that illuminate general mechanisms of function and dysfunction in the human context.

Non-coding variants are typically modest in their effects, so the importance of the underlying pathway cannot be extrapolated from odds ratio data or other measures of variant-attributable risk[37,38]. Importantly, such regulatory variants may highlight pathways for which informative coding variants are not available, either because they would be lethal or because their effects are so florid as to obscure more subtle phenotypes, such as altered risk for polygenic autoimmunity. Mutations in *SMCHD1* present with bosma arhinia microphthalmia syndrome (BAMS) or facioscapulohumeral muscular dystrophy type 2 (FSHD2), phenotypes not associated with RA or T1D[39,40]. Patients lacking *ICOS* develop common variable immunodeficiency, a much broader form of immune dysfunction[41]. It is therefore unlikely that exploration of rare coding variants could have uncovered the SMCHD1-ICOS-Tph axis identified here.

Sequential application of FREP and mass spectrometry allowed us to identify SMCHD1 as a regulatory protein mediating the effect of rs117701653 in T lymphocytes. SMCHD1 consists of an N-terminal ATPase domain and a C-terminal SMC hinge domain that binds directly to DNA. The hinge domain highly binds to poly-dC ssDNA but not poly-dA in vitro, despite limited sequence-specific binding[25], a biology reflected in our data in the loss of binding to the rs117701653 A allele

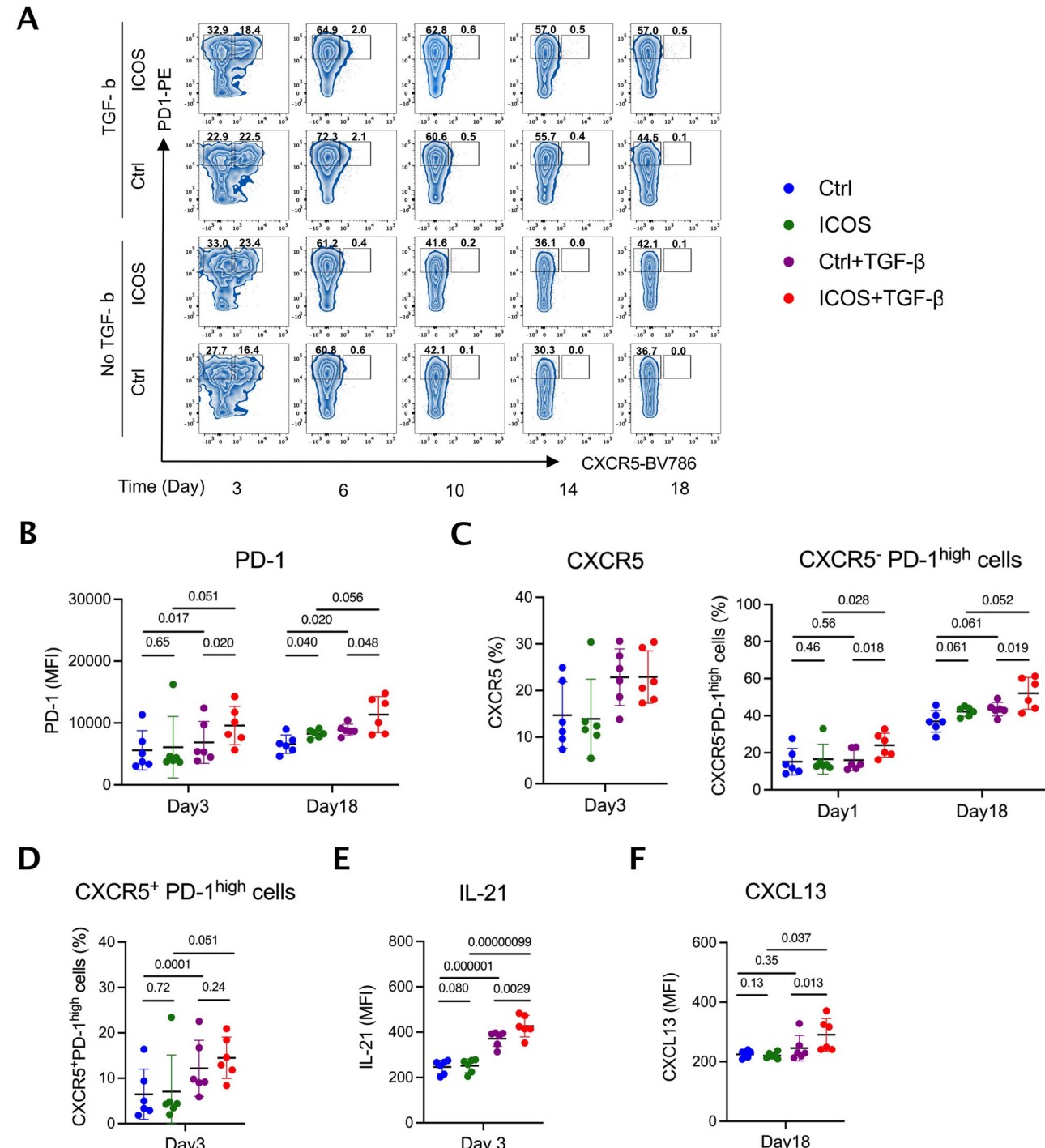

**Fig. 5 | ICOS stimulation accelerates development of TGF-β-induced CXCR5⁻ PD-1^high Tph-like cells expressing IL-21 and CXCL13.** Human memory CD4⁺ T cells from 6 healthy subjects were differentiated with anti-CD3/CD28 stimulation in the indicated combination of TGF-β and anti-ICOS stimulation. **A** Representative dot plots by flow cytometry of CXCR5 and PD-1 to gate CXCR5⁻PD-1^high Tph-like cells and CXCR5⁺PD-1^high Tfh-like cells. Surface CXCR5 and PD-1 expression, and intracellular IL-21 and CXCL13 expression to detect Tph cells were evaluated in Control (blue), ICOS (green), TGF-β (purple), TGF-β + ICOS (red) stimulation groups. For each group consisting of n = 6 healthy donors, (**B**) MFI of PD-1 in whole population, (**C**) frequency of CXCR5⁺ and CXCR5⁻ PD-1^high cells, (**D**) frequency of CXCR5⁺ PD-1^high cells, MFI of (**E**) IL-21 and (**F**) CXCL13 are shown after 3 days or 18 days of differentiation (Supplementary Fig. 10; results at all times tested). MFI, mean fluorescence intensity. Error bars are mean ± S.D. *P < 0.05, **P < 0.01, ***P < 0.001, ****P < 0.0001. P values determined using one-way ANOVA with uncorrected Fisher's LSD test.

when the nuclear extract was pre-incubated with antibodies against the SMCHD1 hinge. SMCHD1 controls long-range chromatin repression, potentially though the formation of chromatin loops that may impede promoter-enhancer interactions[42,43]. This capacity to serve as a transcriptional insulator may explain the capacity of

rs117701653 to regulate *ICOS* rather than the more proximate genes *CD28* and *CTLA4*.

Recent genome-wide immune trait association studies in healthy individuals find that autoimmunity-associated loci can control the frequency of circulating innate and adaptive immune cells, cell surface

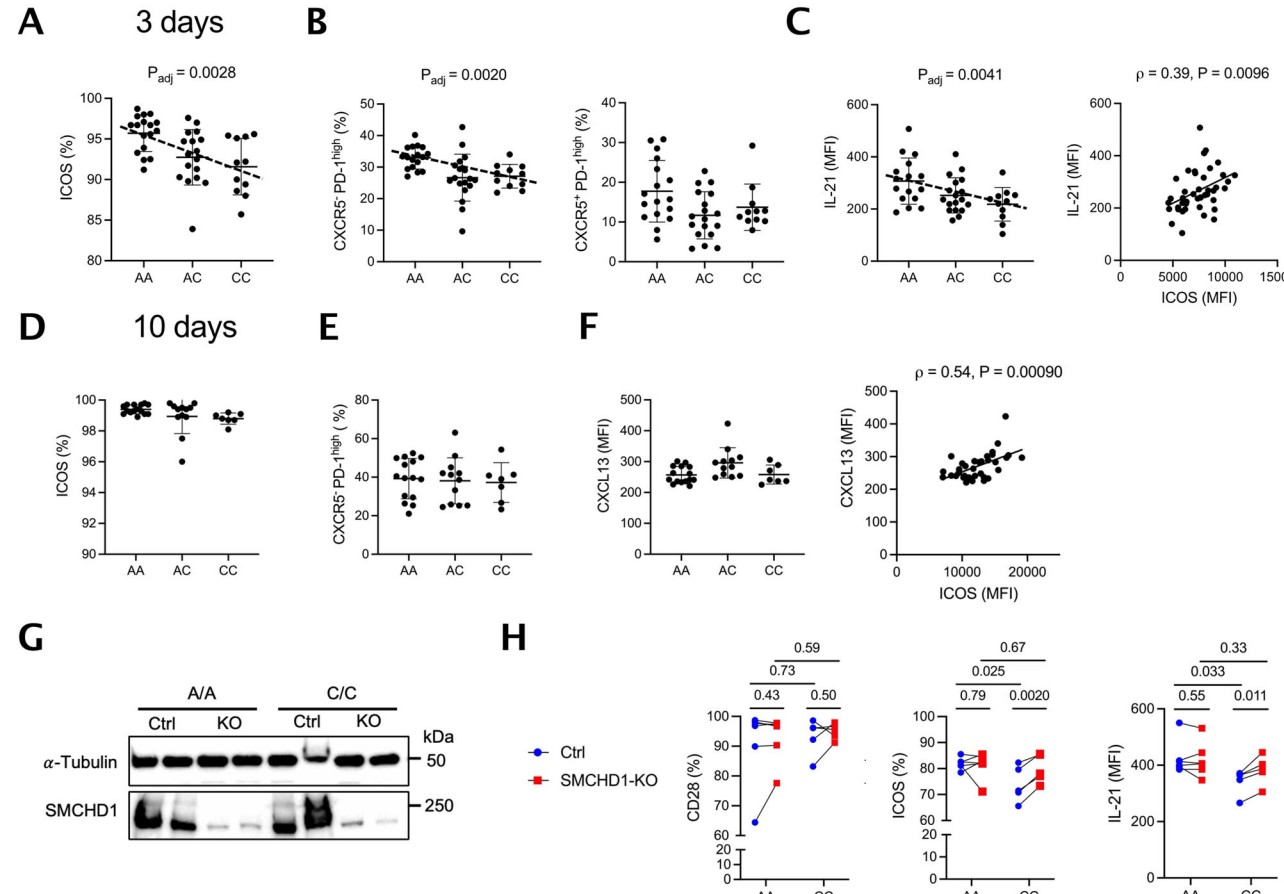

**Fig. 6 | rs117701653-SMCHD1 accelerates differentiation of CXCR5⁻PD-1^high Tph-like CD4⁺ T cells.** Memory CD4⁺ T cells from 46 healthy donors with A/A ($n$ = 17), A/C ($n$ = 18), C/C ($n$ = 11) genotype at rs117701653 were differentiated using anti-CD3/CD28 beads and anti-ICOS in the presence of TGF-β. After 3 days of differentiation, (**A**) frequency of ICOS⁺ cells, (**B**) frequency of CXCR5⁻PD-1^high and CXCR5⁺PD-1^high cells, (**C**) MFI of IL-21 and correlation between MFI of ICOS and IL-21 across genotypes. After 10 days of differentiation in 34 donors, A/A ($n$ = 15), A/C ($n$ = 12), C/C ($n$ = 7), (**D**) frequency of ICOS⁺ cells, (**E**) frequency of CXCR5⁻PD-1^high cells, (**F**) MFI of CXCL13 and correlation between MFI of ICOS and CXCL13 across individuals. **G, H** Memory CD4 + T cells from A/A ($n$ = 6) and C/C ($n$ = 5) donors were preactivated with anti-CD3/CD28 for 2 days, nucleofected with either non-targeting or *SMCHD1*-targeting sgRNA through CRISPR-Cas9, and subsequently differentiated with anti-CD3/CD28, TGF-β and anti-ICOS stimulation. After 3 days of differentiation, (**G**) SMCHD1 protein level was analyzed by western blot in the negative control and SMCHD1-deleted groups for 2 A/A and 2 C/C donors, and (**H**) expression of CD28, ICOS, and IL-21 was measured by flow cytometry in 11 donors, A/A ($n$ = 6) and C/C ($n$ = 5). SMCHD1 expression in both control (blue) and SMCHD1-deleted (red) cells was independently confirmed by western blot in two replicates (Supplementary Fig. 13). MFI, mean fluorescence intensity. Error bars are Mean ± S.D. *P* values determined using a linear regression model adjusted for age and sex (**A**–**C**) or paired t-test with two-tailed significance (**H**). Pearson correlation with a two-tailed test (**C, F**). Uncropped western blots are included in the Source Data file.

marker expression, and cytokine production[30,31,44]. To date, published studies have failed to identify any immune phenotypes associated with rs117701653, likely due to strict genome-wide significance penalties and limited power for the low-frequency C variant, again illustrating the essential role for multiple complementary approaches to uncover relationships between genotype and immune traits.

Discovered only recently, Tph cells play an important role in B cell antibody production beyond the germinal center, where Tfh cells fulfil this role[21,45]. Avoiding recruitment to lymphoid organs through the absence of CXCR5, Tph cells elaborate CXCL13 to drive B cell recruitment and IL-21 to promote B cell differentiation into plasmablasts. In RA, they are abundant in inflamed joints from patients with rheumatoid factor or ACPA – so-called seropositive RA – whereas patients with seronegative arthritis typically exhibit far fewer Tph cells, likely reflecting at least in part the pathogenic relevance of immune complexes in seropositive disease[21,46]. Tph cells have also been implicated in other autoantibody-associated diseases, including T1D[22]. Our findings show that ICOS accelerates the acquisition of the CXCR5⁻PD-1^high Tph phenotype and amplifies both CXCL13 and IL-21 production. TGF-β remains critical to the development of the phenotype[35,47],

although the important auxiliary role of ICOS in vivo is evident from the clear association between ICOS expression (and its driver rs117701653 genotype) and Tph abundance in blood and inflamed joints. Further study will be required to delineate the mechanism by which TGF-β and ICOS work together to develop and modulate the Tph phenotype.

As noted, the protective C allele of rs117701653 is relatively uncommon, and although an impact of heterozygous carriage was observable, the most striking impact on ICOS expression and Tph abundance was noted in C/C homozygotes. Such individuals are uncommon, representing 1% or less of many populations, raising the possibility of purifying selection as has been observed in other loci[48–50]. Indeed, recent evidence suggest that the *CTLA4/ICOS* locus was subjected to selection pressure by the Black Death[51]. Our data do not directly address this possibility. However, no deviation from Hardy-Weinberg equilibrium is evident for rs117701653 in any known population (Supplementary Table 2).

Importantly, our observation that rs117701653 drives ICOS expression and Tph abundance was derived in healthy donors rather than in patients with RA or T1D. This was necessary because GWAS

identify variants that modulate disease risk, that is, variants that alter the probability that a healthy individual will develop disease. Genetic variants have been described that have different or even opposing effects in healthy and disease contexts[52]. However, only variants active in healthy donors can become GWAS hits, though it remains possible that such variants might also impact disease severity, a possibility not explored here for rs117701653.

There remain several limitations to our study. rs117701653 is one of two independent SNPs implicated by fine mapping at the CD28/CTLA4/ICOS locus[15]. The gene target of the second SNP, rs3087243, remains to be determined. Our data do not exclude the possibility that additional functional SNPs reside in tight LD with rs117701653. For example, the *STAT4* locus contains two SNPs within a single haplotype that each modulate gene expression through the binding of distinct transcription factors[53]. However, at present no data suggest a similar process for rs117701653. Importantly, altered binding of SMCHD1 to rs117701653 could modulate genes beyond *ICOS*, and variation in *ICOS* will likely affect immune functions beyond Tph cell abundance. For example, ICOS enhances the development of Tfh cells, consistent with the trends observed here between memory CD4 + T cell ICOS expression and Tfh abundance[20]. Our data, focused on CD4$^+$ T cells in a narrow range of conditions, are insufficient to conclude that Tph abundance represents the mechanism through which rs117701653 drives disease. Rather, the net impact of rs117701653 on risk for RA and T1D will represent the sum of all effects on all lineages and conditions, of which enhanced Tph abundance likely represents only one component.

Taken together, our data identify a genetic mechanism, implicated in risk for RA and T1D, through which ICOS regulates the differentiation of Tph cells. These studies model an approach for understanding non-coding functional variants and highlight the importance of regulatory variants as a tool to understand susceptibility to common polygenic diseases.

## Methods

### Human samples
This research complies with all relevant ethical regulations, as approved by the Institutional Review Board at Mass General Brigham. Written informed consent was obtained from all participants, except as noted below. Blood samples from healthy subjects were recruited from genotyped volunteer donors within the Mass General Brigham Biobank through the Recruitment Core of the Joint Biology Consortium (JBC, www.jbcwebportal.org). 24 subjects (8 A/A, 8 A/C, and 8 C/C genotypes at rs117701653) were enrolled for low-input RNA sequencing. 46 subjects (17 A/A, 18 A/C, 11 C/C), including 7 individuals (2 A/A, 3 A/C, 2 C/C) included also among the first cohort of 24 subjects, were recruited for CD4$^+$ memory T cell immunophenotyping and differentiation assays.

Patients with RA fulfilled the ACR/EULAR 2010 Rheumatoid Arthritis classification criteria[54]. Synovial fluid samples were obtained as excess material from patients undergoing clinically indicated diagnostic or therapeutic arthrocentesis as directed by the treating rheumatologist. Blood samples from healthy controls were obtained from blood bank leukoreduction collars from anonymous platelet donors, and as de-identified discard samples, were obtained without written consent and were not employed for genomic studies. Synovial fluid samples from RA patients and blood samples from healthy controls were used to estimate the correlation between ICOS expression and Tph proportion in memory CD4$^+$ T cells.

### FREP and mass spectrometry
FREP followed the previously described protocol[23]. The bait DNA fragment (rs117701653-C/5Biosg), competitor DNA fragment (rs117701653-C), and irrelevant DNA sequence were used; sequences are listed in Supplementary Table 3. Mass spectrometry was

performed using a Thermo Scientific Q Exactive HF Orbitrap LC-MS/Ms system.

### Electrophoretic mobility shift assay (EMSA)
EMSA was performed using the LightShift Chemiluminescent EMSA kit (Thermo Scientific, 20148) according to manufacturer's instructions. A probe made of the 31-bp sequences centered on SNP rs117701653 was made by annealing two biotinylated oligonucleotides. Nuclear proteins were extracted from Jurkat T cells using NE-PER Nuclear and Cytoplasmic Extraction Reagents (Thermo Scientific, 78835) per manufacturer's instructions. For gel supershift, the indicated antibody was added before or after an additional 30 min incubation with DNA probe and nuclear protein extract (Supplementary Fig. 1).

### Chromatin Immunoprecipitation (ChIP) qPCR
ChIP-IT PBMC kit (Catalog no. 53042, Active Motif) was used according to the manufacturer's instructions. In brief, human CD4 + T cells or CRISPR-Cas9-edited Jurkat cell clones were cross-linked for 15 min using 1% formaldehyde. Cross-linked cells were lysed, and chromatin was sheared by using a QSONICA Q125 (42% amplitude, pulse 30 seconds on/off for 5 minutes of "on" time per rounds, total four rounds of sonication). 20 μg of chromatin was incubated with either 10 μg of anti-SMCHD1 (Abcam, ab179456) or 10 μg of rabbit IgG (NOVUS, NB810-56910) to control for non-specific binding. Quantitative PCR was used to measure relative fold enrichment of SMCHD1 binding at SNP rs117701653 or HS17 promoter as a positive control[55]. ChIP-qPCR primers are listed in Supplementary Table 3.

### HDR-editing and SMCHD1 deletion by CRISPR-Cas9
CRISPR-mediated homology-directed repair (HDR) was applied for the generation of HDR-edited Jurkat cell lines using sgRNA targeting three bases upstream from rs117701653 and asymmetrical single-stranded DNA donors[29]. $2 \times 10^5$ Jurkat cells were nucleofected with 20 picomole of sgRNA-Cas9 complex and 100 picomole of DNA donor template using program CL-120 of Amaxa™ 4D-Nucleofector and SE cell line kit S (Lonza, V4XC-1032). The edited single-cell clones were sorted into 96-well plate by BD Aria II sorter and expanded for two months. Using a GeneArt™ Genomic Cleavage Detection Kit (ThermoFisher Scientific, A24372) following manufacturer's instructions, modified DNA was isolated from the survived clones (904/1,440) and used in a PCR reaction. PCR products were analyzed by Sanger sequencing to identify A/A wild-type clones ($n = 38$), A/C edited clones ($n = 3$), C/C edited clones ($n = 3$).

For generation of SMCHD1 deleted A/A and C/C Jurkat clones, $2 \times 10^5$ cells were nucleofected with 20 picomole of sgRNA-Cas9 complex that targets exon 8 of the *SMCHD1* gene using program CL-120 of Amaxa™ 4D-Nucleofector and SE cell line kit S. Afterward, the cells were cultured for 4-5 days and analyzed for SMCHD1 deletion by western blot. For generation of SMCHD1 deleted A/A and C/C differentiated Tph cells, memory CD4 + T cells from 6 A/A and 6 C/C healthy donors were preactivated with anti-CD3/CD28 beads for 2 days. Thereafter, $1 \times 10^6$ cells were nucleofected with either non-targeting or SMCHD1 targeting sgRNA using a 40 picomole of sgRNA-Cas9 complex. The nucleofection was performed using program EO-115 of Amaxa™ 4D-Nucleofector and P3 primary cell kit S. The nucleofected cells were then differentiated with anti-CD3/CD28 beads, TGF-β and anti-ICOS stimulation for 3 days. Deleted SMCHD1 was confirmed by western blotting. IL-21 and ICOS expression in SMCHD1-deleted Tph cells were compared to control cells using flow cytometry. The sequences of sgRNAs and DNA donor template are listed in Supplementary Table 3.

### Protein and RNA quantifications in Jurkat clone cells
Protein levels in resting and stimulated Jurkat clones were measured by western blotting. A/A and C/C clones were stimulated with either plate-

bound anti-CD3 (Biolegend, 317302)/CD28 (Biolegend, 302943) or anti-CD3/ICOS (Invitrogen, 16-9948-82) antibodies over different time periods. Whole-cell lysate were obtained using Cell Lysis Buffer (Cellsignal, 9803). Protein extracts were subjected to western blotting using anti-SMCHD1 (ABCAM, ab179456), anti-$\alpha$-Tubulin (Cellsignal, 2144 S), anti-ICOS (ABCAM, ab175401), anti-RAPH1 (Cellsignal, 91138 T), anti-CD28 (Cellsignal, 38774 S), anti-AKT (Cellsignal, 9272 S), Ser473 Phospho-AKT (Cellsignal, 4060 S), JNK (Cellsignal, 9252 S), Thr183/Tyr185 Phospho-JNK (Cellsignal, 4668 S) antibodies. Details of the antibodies are available in Supplementary Table 4.

For measurement of RNA level, total RNA was isolated from resting Jurkat clones using the RNeasy Micro kit (Qiagen, 74004) and used for cDNA library construction using oligo-dT primer and reverse transcriptase (Agilent, 600559). Quantitative PCR was performed with SYBR green fluorescent dye (Agilent, 600882) using the real-time qPCR detection system (ThermoFisher Scientific, QuantStudio3). We used specific primers from PrimerBank (https://pga.mgh.harvard.edu/primerbank/index.html) for human *ACTB*, *SMCHD1*, *ICOS*, *CD28*, and *RAPH1*. Relative mRNA levels were estimated using the comparative Ct method, $\Delta\Delta$Ct method[56]. All qPCR primers are listed in Supplementary Table 3.

### PBMC isolation and immunophenotyping

Whole blood was collected from 46 healthy human subjects based on genotype at rs117701653. Peripheral blood mononuclear cells (PBMCs) were isolated by a Ficoll gradient (GE Healthcare, 17-1440-02) and cryopreserved in 10 % DMSO in fetal bovine serum. Immunophenotyping was carried out by flow cytometry on thawed PBMCs. We characterized major populations of human CD4+ T cells as follows: naïve T cells (CD3+CD4+CD45RA+), memory T cells (CD3+CD4+ CD45RA−), and effector memory subsets: Th1 (CD3+CD4+CD45RA− CCR6−CXCR3+CCR4−), Th2 (CD3+CD4+CD45RA−CCR6−CXCR3−CCR4+), Th17 (CD3+CD4+CD45RA−CCR6+CXCR3−CCR4+), memory Treg (CD3+CD4+CD45RA−CD25highCD127−FOXP3+), Tfh (CD3+CD4+CD45RA−CXCR5+PD-1high), and Tph (CD3+CD4+CD45RA−CXCR5−PD-1high) (Supplementary Fig. 5). Fluorescent-conjugated and isotype control antibodies used in flow cytometry are listed in Supplementary Table 4.

For gating Tph and Tfh cells, we considered cells with a higher signal level than the negative controls as positive for CXCR5 and PD-1 expression, and cells with similar signal to the control as negative. To distinguish cells with high and intermediate PD-1 among the positive cells in resting T cells, we established a threshold of 4,100 on a biexponential scale (Fig. 3C). For the differentiated Tph cells, we used a threshold of 9,700 on a biexponential scale to clearly distinguish cells with PD-1 high expression after 3 days of differentiation (Fig. 5A).

### RNA-Sequencing and data processing

CD3+CD4+ T cells from PBMCs of healthy subjects from the Mass General Brigham Biobank genotyped for rs117701653 (A/A = 8, A/C = 8, C/C = 8) were isolated by negative selection using EasySep human CD4+ T cell Isolation kit (STEMCELL, 17952). From the isolated CD3+CD4+ T cells of each individual, RNA was isolated using RNeasy Micro kit (Qiagen, 74004) and eluted in 14 μl of water. 10 ng samples of RNA were transferred into wells of a 96-well plate, and RNA-seq libraries were prepared at Broad Technology Labs at the Broad Institute of MIT and Harvard (Cambridge, Massachusetts, USA) using the Illumina SmartSeq2 platform. Samples were sequenced on a NextSeq 500 generating a median of 5.6 million 38 bp paired-end reads per sample.

Raw data were processed using release 3.9 of the nextflow nf-core "rnaseq" pipeline[57,58]. The pipeline was executed on the BCH HPC Clusters Enkefalos 2, using singularity[59] containers to ensure optimal reproducibility. Briefly, we performed adapter and quality trimming using Trim Galore (version 0.6.7) and subsequently aligned reads to the GRCh38 reference genome using STAR[60] (version 2.7.10). We quantified transcript expression with Salmon[61] (version 1.5.2) and aggregated transcript abundances to gene-level measurements with bioconductor-tximeta[62] (version 1.8.0).

### Targeted eQTL analysis

We used QTLtools[63–65] (version 1.3.1-12-gba66d62ef4) to perform a targeted cis-eQTL mapping analysis for protein coding genes that were expressed above 0 log$_2$(tpm+1) in at least 8 samples and have a transcription start site (TSS) within a 1MB window of SNP rs117701653, namely *WDR12*, *NBEAL1*, *CYP20A1*, *ABI2*, *RAPH1*, *CD28*, *CTLA4*, *ICOS*, and *PARD3B*. We corrected expression levels for age and sex and ranknormal transformed residuals via QTLtools' "--normal" option. We computed nominal *P*-values for the association between genotypes and expression levels of the selected genes using a linear model implemented in QTLtools.

### Memory and naive CD4+ T cell isolation and differentiation

Memory and naive CD4+ T cells from PBMCs of healthy subjects were isolated by negative selection using the EasySep human memory CD4+ T cell Isolation kit (STEMCELL, 19157) and the human naive CD4 + T cell isolation kit II (Miltenyi, 130-094-131). T cells were resuspended in RPMI (supplemented with 10 % fetal bovine serum and 100 units/mL penicillin/streptomycin) at 0.25 × 10$^6$ cells/mL and stimulated with anti-CD3/CD28 beads (Invitrogen, Dynabeads human T-activator CD3/CD28) at a ratio of 5:1 (cell:bead), to which the indicated combination of 2 ng/mL TGF-β1 recombinant protein (R&D system, 7754BH005/CF) and 2 μg/ml anti-ICOS antibody (Invitrogen, 16-9948-82) were added for varying periods (3, 6, 10, 14, and 18 days). For the prolonged expansion, cells were stimulated again with anti-CD3/CD28 beads together with TGF-β, ICOS stimulation by anti-ICOS, or both at 6, 10, and 14 days of differentiation.

Cells were harvested at indicated time points for intracellular cytokine staining. Cells were re-stimulated with anti-CD3/CD28 beads at a ratio 5:1 (cell:bead) for 24 h and with both phorbol 12-myristate 13-acetate and ionomycin (Both 1:500, Biolegend 423301) for the last 5 hours. Brefeldin (1:1000, BD Bioscience 555029) and monensin (1:1500, BD Bioscience 554724) were added for the last 5 hours. Cells were washed twice in cold PBS, incubated for 30 minutes with Fixable Viability Dye (Invitrogen, 65-0863-14), washed in 1% FBS/PBS, and then incubated in cell surface antibodies with anti-CD3, CD4, PD-1, CXCR5 for 20 minutes. Cells were then washed again in 1% FBS/PBS, and fixed and permeabilized using Transcription Factor Buffer Set (eBioscience, 00-5523-00). Permeabilized cells were incubated in intracellular antibodies with anti-IL-21 and CXCL13 for 1 hour. Flow cytometry analysis was performed on a BD Fortessa analyzer. Antibodies used in flow cytometry are listed in Supplementary Table 4.

### Mass cytometry data

AMP mass cytometry data followed the previously described gating method[34]. Briefly, frequency of Tph cells (CD3+CD4+CD45RO+CXCR5−PD-1high), Tfh cells (CD3+CD4+CD45RO+CXCR5+PD-1high), plasmablasts (CD45+CD19 + CD20−CD38highCD27+) were quantified by manual gating, with uniform gates applied to all samples. FlowJo 10.8.0 was used for determination of cell population frequencies.

### Reporting summary

Further information on research design is available in the Nature Portfolio Reporting Summary linked to this article.

## Data availability

Source data are provided with this paper. All data supporting the findings of this study are available within the paper and its Supplementary Information. Raw data for each panel may also be accessed through Nigrovic, Peter (2023), "2023Kim-ICOS-Tph", Mendeley Data, V1, https://doi.org/10.17632/7263bjmtxd.1[66]. The mass spectrometry

protemics data have been deposited to the ProteomeXchange Consortium via the PRIDE[67] partner repository (https://www.ebi.ac.uk/pride/) with the dataset identifier PXD048977 and 10.6019/PXD048977. The raw RNA sequencing data generated in this study are available through the database of Genotypes and Phenotypes (dbGaP) under accession code phs003448.v1.p1. Access to this controlled data is intended to be consistent with the research participants' informed consent and to ensure the confidentiality and privacy of participants. Permanent employees of an institution at a level equivalent to a tenure-track professor or senior scientist with laboratory administration and oversight responsibilities may request access through dbGAP. Interested parties may request access to the data by submitting a Data Access Request (DAR) through the dbGaP website (https://dbgap.ncbi.nlm.nih.gov/aa/wga.cgi?page=login). Access to the data is subject to approval by the Data Access Committee (DAC) and is contingent upon compliance with Data Use Certification (DUC) Agreement. Approved Users are permitted to use the data solely for the specific research project described in the Data Access Request (DAR). Typically, the request takes 14-21 days for approval and access is permitted for 12 months. Gene expression matrices are accessible in NCBI's Gene Expression Omnibus via GEO Series accession number GSE235868. Source data are provided with this paper.

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

## Acknowledgements

TK, M-MB and QW were supported by Joint Biology Consortium microgrants off parent grant NIH/NIAMS P30AR070253. NH is supported by an MD fellowship from the Boehringer Ingelheim Fonds. JAS was supported by NIH/NIAMS R01AR077607, P30AR070253, and P30AR072577, the R. Bruce and Joan M. Mickey Research Scholar Fund, and the Llura Gund Award for Rheumatoid Arthritis Research and Care. The project described was supported by Clinical Translational Science Award 1UL1TR002541-01 to Harvard University and Brigham and Women's Hospital from the National Center for Research Resources. Dr. Sparks has received research support from Bristol Myers Squibb and performed consultancy for AbbVie, Amgen, Boehringer Ingelheim, Bristol Myers Squibb, Gilead, Inova Diagnostics, Janssen, Optum, and Pfizer unrelated to this work. YB was supported by a National Institutes of Health grant (R01AR063759), an ASPIRE an investigator-led award from Pfizer, and by the Broad Institute. B-HK was supported by NIH grant T32AR007530-35. RD was supported by the Arthritis National Research Foundation and NIH/NIAMS P30AR070253. M-GA was supported by NIH/NIAMS P30AR070253, Gilead Sciences, Lupus Research Alliance, and the Arthritis National Science Foundation Vic Braden Family Fellowship. The author would like to acknowledge Boston Children's Hospital's High-Performance Computing Resources BCH HPC Clusters Enkefalos 2 (E2), and Massachusetts Green High-Performance Computing (MGHPCC) made available for conducting the research reported in this publication. Software used in the project was installed and configured by BioGrids[68]. MTW was supported by NIH grants R01NS099068, R01HG010730, U01AI130830, R01AI024717, and R01AR073228. SR was supported by 5R01AR063759-07. DAR was supported by NIH/NIAMS P30AR070253, K08AR072791, R01AR078769, and a Burroughs Wellcome Career Award in Medical Sciences. PAN was supported by investigator-initiated grant IM101-835 from BMS, NIH/NIAMS 2R01AR065538, R01AR075906, R01AR073201, P30AR070253, the Fundación Bechara, and the Arbuckle Family Fund for Arthritis Research. We thank Dr. Diogo Meyer (University of Sao Paulo) for helpful discussions on population genetics analyses.

## Author contributions

TK, MM-B, PAN conceived and designed the study. TK, M-MB, QW, NH, YB, BK, RD, RL-B, XC, VRCA, DJC, MG-A, H-JW, MTW, SR performed experiments and/or data analysis and interpretation. JAS led recruitment of genotyped human subjects. DAR advised on Tph experiments. TK, PAN drafted the manuscript. All authors edited and approved the manuscript. PAN supervised the research.

## Competing interests

The authors declare no competing interests.
