## [Peer Review File · Nature Communications]

Non-coding autoimmune risk variant defines role for ICOS in T peripheral helper cell developmentREVIEWER COMMENTS

Reviewer #1 (Remarks to the Author):

This study reports several significant new findings including the fact that the rs117701653 SNP associates with the % of ICOS expression in CD4+ T cells and its protective allele binds to SMCHD1 which represses the expression of ICOS. The same allele associates with a lower % Tph in healthy subjects. Finally the authors show that ICOS stimulation has an adjuvant effect on the differentiation of Tph cells in vitro and that the protective allele of the rs117701653 SNP inhibits the differentiation of Tph in this assay. Some of the differences shown in the study are a bit marginal (mainly Fig. 5) and some Tph vs Tfh claims are perhaps a bit overstated (e.g. Tfh seem to show the same trend in Fig. 4h and with a bit more points the correlation would become significant), but overall this work is original and convincing.

A few aspects that might deserve a bit more attention:

1. The work on Jurkat clones clearly shows that SMCHD1 specifically regulates ICOS but genome editing in Jurkats requires multiple passaging which makes comparisons of gene expression between clones a bit delicate. It is recommended that complementary experiments where knockdown of SMCHD1 in T cells from C/C carriers compared to A/A cells are performed. It is recommended that at least ICOS expression is studied although best would be to show that knockdown of SMCHD1 differentially affects the assay show in Fig. 5 and 6.
2. There does not seem to be an association with the resting levels of CD28 in primary T cells but the Jurkat clones show a trend of decreased expression of CD28. Does the rs117701653 SNP associate with CD28 expression in stimulated T cells from healthy subjects or T cells from RA patientsA? Are there differences in ICOS vs CD28 signaling in A/A vs C/C Jurkat clones or stimulated genotyped primary T cells?
3. It would be good to at least discuss any idea of how SMCHD1 represses ICOS expression. Is anything other protein recruited to the SMCHD1 complex once it binds to the C allele of the rs117701653 SNP? Also the fact that the SNP is so far from the ICOS gene leaves a chance that the effect on ICOS might not be the only mechanism through which SMCHD1 binding to the rs117701653 SNP protects from autoimmunity.

Reviewer #2 (Remarks to the Author):

In this paper, Kim et al. investigated the role of the rs117701653 non-coding variant in immune cells. They demonstrated that the risk variant is associated with a decrease binding of the SMCHD1 regulator leading to an increased expression of ICOS thus an increase abundance of particular T-cell subsets.

Overall, this paper is straightforward and easy to follow. The methods are appropriate and described adequately. They use a great variety of models to confirm their hypothesis (in vitro, primary cells from healthy donors and patients, engineered cell lines). This study provides novel data. They manage to pinpoint a strong correlation between a risk variant and immune phenotypes likely to participate in the development of the disease.

Minor comments:

-The figures aren't really "engaging" compared to the manuscript but their content is appropriate and explicit (+need to add Tfh label on figure 3c).

-It would be interesting for the authors to comment in the body of the text about the frequency of the risk variant. The MAF is between 5 and 10% (according to dbSNP) suggesting that the risk variant is present at 90 to 95% in the population. How do they explain such a great effect size in healthy controls for a variant that is so common? Which additional mechanism could actually lead to the disease?

-Is there the same correlation between rs117701653 genotype and the different observed phenotypes (ICOS expression, Tph abundance...) in RA patients compared to healthy donors? And what about T1D?

Reviewer #3 (Remarks to the Author):

This is a very nice study which biochemically delineated the impact of rs117701653. Authors show preferred genotype dependent binding of SMCHD1, and ensuing levels ICOS and Tph frequency. I think this work nicely documents the elusive role for non-coding variants.

Major comments:

1) I think clarity of nomenclature should be addressed. I understand that it can be lower or enhanced risk, but I think title and language should be clear in calling the allele studied as a protective allele first and foremost.

2) Also genetically it should be clearly spelled out that the biggest effect is seen in homozygosity of CC as a protective allele. Homozygosity in all populations per Gnomad is actually very rare, compared to what one could expect from rate of heterozygosity, which suggests some form of purifying selection. Thus

there may have been a price to be paid for this protection (and should be discussed in light of rare genetics known).

3) Figure 1 D. Positive control variance supersedes the actual (despite reasonable and statistically tight) results of rs117701653, which make me wonder if the finding is robust enough. Perhaps it should be confirmed in PHA blasts from rs117701653 carrying individuals.

4) I think the correlations shown in figure 4 are nice, but I don't understand where the connection to rs117701653 is? Do patients with RA or T1D with rs117701653 have more or less Tph cells and all the ensuing downstream effects? Without this it very hard to interpret these results in the context of RA or T1D which was the point of this paper. At this point this rs117701653 is merely correlated and appears to cause with ICOS expression in healthy subjects.

Reviewer #4 (Remarks to the Author):

Kim et al. showed that the protective C allele of rs117701653 is involved in the repression of ICOS expression via binding of SMCHD1. Furthermore, they try to show that rs117701653 is involved in Tph cell differentiation through ICOS. Unfortunately, however, the experiments on differentiation have had procedural and experimental design problems and are insufficient to show its involvement in Tph differentiation. Furthermore, the involvement of SMCHD1 in Tph cell differentiation by rs117701653 has not been addressed. To reach the level of publication, it is required to address the following issues.

#The definitions of positive and negative expression (such as FMO or isotype control) of molecules including ICOS and PD-1 in flow cytometry needs to be specified in gating strategy. If Isotype control antibodies are used in the definition, please add them to the table. Furthermore, a clear definition of PD-1(high), which is crucial for Tph and Tfh cells, needs to be described in this manuscript to ensure the reliability of the Tph and Tfh cell frequency data.

#Fig5 is intended to show that ICOS signaling is important for Tph cell differentiation in vitro. However, the Tph cell differentiation culture conditions in Fig 5 and 6 may not be appropriate. Previous studies have shown that the Tph cell phenotype, PD-1 expression and CXCL13 production, can be induced under TGFβ alone conditions, but this is not reproduced by the conditions in this paper. (Fig 5B, C, and F). Although it is known that about 20% of synovial CD4-positive cells in actual rheumatoid arthritis patients produce CXCL13, the MFI of CXCL13 and IL21 in Fig 5 and S8 is quite low (~300) compared to PD-1 (-100000) and other factors. Please show the positivity rate and a representative dot plot of CXCL13 and IL21 in FigS8. Previous studies have often used plate bound CD3 antibodies and naïve human CD4 T cells for Tph cell differentiation. Although the use of memory CD4 in Fig. 5 to investigate the role of ICOS signaling in Tph cell differentiation is reasonable, CD3 stimulation method needs to be optimized to achieve adequate Tph differentiation.

#Fig S8: PD-1 expression and CXCR5-PD-1(high) frequency are higher with ICOS stimulation on Day 3 and Day 18, but without ICOS stimulation on Day 6. It is expected that the improvement of induction conditions will lead to more consistent results throughout the study period. Is it reasonable to argue based solely on the results of Day 3 and Day 18?

#Fig6: As shown in Fig. 3 for, the frequency of ICOS-positive cells and Tph cells in memory CD4 T cells differ by genotype from the beginning day 0. The Day 3 results only reflect the Day 0 state in short differentiated cultures (Fig. 6A,B) and do not indicate that genotype is involved in the in vitro differentiation of Tph cells. Rather, it is better to culture naive CD4, which initially expresses little ICOS independent of the genotype (Fig 3B), in the in vitro differentiation. By differentiating cells with no difference in initial ICOS expression, it is possible to show that differences in ICOS levels caused by genotyping can affect Tph cell differentiation.

#The involvement of SMCHD1 in Tph cell differentiation by #rs117701653 has not been addressed. Tph cell differentiation using SMCHD1 knockout or lentivirus/siRNA knockdown CD4T cells can be used to demonstrate SMCHD1 involvement.

#There are some discrepancies between the results of Fig2B and Fig2C. For example, in CD4+T, the expression of ICOS in AC is intermediate between in AA and CC, but in Jurkat qPCR, its expression in AC is as decreased as in CC. Furthermore, while there is no difference in CD28 mRNA in CD4, protein level of CD28 in Jurkat shows a decrease in CD28 in CC and p values are lacking. Discussion about these discrepancies is required.

#One important issue of this paper is that ICOS stimulation is more involved in Tph cell differentiation than in Tfh cells, but the reasons for this are not discussed. For example, in Fig. 6S, about half of the Tfh cells from healthy subjects are negative for ICOS. Could this explain why ICOS stimulation differentiates Tph cells preferentially in human?

#Please show the ICOS positivity rate in each fraction including Tph cells and Tfh in RA as well as FigS6.

Dear Editors,

We are very grateful for the comments of the reviewers and the opportunity to submit an improved manuscript for reconsideration. We reproduce the comments below in full, together without our responses. We believe we have been able to address all comments and are pleased to return a clarified and strengthened manuscript.

REVIEWER COMMENTS

Reviewer #1 (Remarks to the Author):

This study reports several significant new findings including the fact that the rs117701653 SNP associates with the % of ICOS expression in CD4+ T cells and its protective allele binds to SMCHD1 which represses the expression of ICOS. The same allele associates with a lower % Tph in healthy subjects. Finally the authors show that ICOS stimulation has an adjuvant effect on the differentiation of Tph cells in vitro and that the protective allele of the rs117701653 SNP inhibits the differentiation of Tph in this assay. Some of the differences shown in the study are a bit marginal (mainly Fig. 5) and some Tph vs Tfh claims are perhaps a bit overstated (e.g. Tfh seem to show the same trend in Fig. 4h and with a bit more points the correlation would become significant), but overall this work is original and convincing.

RESPONSE: We are grateful for this positive summary. We agree that the effects observed are sometimes modest, as expected for common non-coding variants; however, impact at the population level is clear from the emergence of this locus in GWAS. We have sought throughout to avoid overstatement and have carefully reviewed the text accordingly. We agree that the Tfh finding trends similarly to the Tph finding, and indeed we had highlighted this in the original Discussion: "For example, ICOS enhances the development of Tfh cells, consistent with the trends observed here between memory CD4+ T cell ICOS expression and Tfh abundance."

A few aspects that might deserve a bit more attention:

1. The work on Jurkat clones clearly shows that SMCHD1 specifically regulates ICOS but genome editing in Jurkats requires multiple passaging which makes comparisons of gene expression between clones a bit delicate. It is recommended that complementary experiments where knockdown of SMCHD1 in T cells from C/C carriers compared to A/A cells are performed. It is recommended that at least ICOS expression is studied although best would be to show that knockdown of SMCHD1 differentially affects the assay shown in Fig. 5 and 6.

RESPONSE: We agree that clone-to-clone variation risks the generation of spurious findings. This problem is generally addressed through multiple independent clones, reasoning that phenotypes arising through random drift are unlikely to be shared among clones. That is the strategy we employed here, and the consistency we observed lends confidence to the hypothesis, especially in the setting of independent corroborative evidence. However, to provide additional support, we have deleted SMCHD1 in memory T cells obtained from A/A and C/C healthy donors and differentiated them into Tph cells. We observed an increased expression of ICOS and IL-21 in C/C donors, consistent with the results obtained from base-edited Jurkat clones. These findings further enhance the robustness of our conclusions that SMCHD1 binding to the C allele represses ICOS expression and Tph differentiation. We have incorporated the updated results in Figure 6G and 6H.

2. There does not seem to be an association with the resting levels of CD28 in primary T cells but the Jurkat clones show a trend of decreased expression of CD28. Does the rs117701653 SNP associate with CD28 expression in stimulated T cells from healthy subjects or T cells from RA patients? Are there differences in ICOS vs CD28 signaling in A/A vs C/C Jurkat clones or stimulated genotyped primary T cells?

RESPONSE: We did not observe a significant effect of the rs117701653 SNP genotype on CD28 levels in resting primary T cells ($p=0.20$ by linear regression) or in edited Jurkat clones (A/A vs. C/C clones RNA level; $p=0.40$, protein level; $p=0.061$) (Fig. 2B,C). However, we acknowledge that ICOS expression trends lower in C/C Jurkat clones.

To investigate the potential association between genetic variation and CD28, we compared the phosphorylation status of AKT (mainly regulated by T cell receptor and ICOS) and JNK (mainly regulated by T cell receptor and CD28) in A/A and C/C Jurkat clones. In line with our expectations, C/C clones exhibiting lower ICOS expression demonstrated a decrease in phosphorylated AKT level upon ICOS stimulation. However, differences were not observed in phosphorylated AKT or JNK levels upon CD28 stimulation (**new Supplementary Fig. 3A,B**). These findings provide additional evidence that the rs117701653 SNP regulates not only ICOS expression but also its intracellular signaling pathways.

While we have the capacity to recruit healthy donors by genotype, we do not have a parallel capacity for RA patients, in whom interpretation of results would in any case be complicated by treatment and systemic inflammation. We note however that the manuscript seeks to define a genetic variant associated with an elevated risk for RA (i.e. for conversion of healthy individuals to ones with RA), and to use this variant as an “experiment of nature” to define novel immunobiology, goals that can be fully accomplished in healthy donors as illustrated by our results.

3. It would be good to at least discuss any idea of how SMCHD1 represses ICOS expression. Is anything other protein recruited to the SMCHD1 complex once it binds to the C allele of the rs117701653 SNP? Also the fact that the SNP is so far from the ICOS gene leaves a chance that the effect on ICOS might not be the only mechanism through which SMCHD1 binding to the rs117701653 SNP protects from autoimmunity.

RESPONSE: We thank the Reviewer for these suggestions. We do not have any data testing the possibility of a protein complex nucleated by the SMCHD1/rs117701653 interaction. At present, the literature offers limited insights into the precise molecular mechanisms and direct protein partners implicated in the long-range repressive chromatin structures by SMCHD1.

However, a proposed insulating model postulates that homodimerized SMCHD1 may contribute to the formation of chromatin loops, potentially impeding promoter-enhancer interactions for adjacent genes.¹ Based on the premise, we assume that the direct binding of SMCHD1 to rs117701653 C allele might interrupt normal interaction between enhancer and ICOS promoter regions, thereby inhibiting *ICOS* gene transcription.

Our study focused on the *ICOS* gene, assessing the effect of rs117701653-SMCHD1 interaction on 11 protein coding genes within a 1 MB window of rs117701653 by eQTL and CRISPR base-editing approaches. We acknowledge that we cannot exclude the possibility of effects extending beyond *ICOS*, given that the chromatin interactions can cover distances from a few kilobases to several megabases.²

We have edited the manuscript to incorporate these considerations and limitations in the discussion section.

Reviewer #2 (Remarks to the Author):

In this paper, Kim et al. investigated the role of the rs117701653 non-coding variant in immune cells. They demonstrated that the risk variant is associated with a decrease binding of the SMCHD1 regulator leading to an increased expression of ICOS thus an increase abundance of particular T-cell subsets.

Overall, this paper is straightforward and easy to follow. The methods are appropriate and described adequately. They use a great variety of models to confirm their hypothesis (in vitro, primary cells from healthy donors and patients, engineered cell lines). This study provides novel data. They manage to pinpoint a strong correlation between a risk variant and immune phenotypes likely to participate in the development of the disease.

RESPONSE: We are grateful for this very positive summary.

Minor comments:

-The figures aren't really "engaging" compared to the manuscript but their content is appropriate and explicit (+need to add Tfh label on figure 3c).

RESPONSE: We thank the Reviewer for this comment and have sought to have the text narrate clearly, through recognize that the experiments from which the data are less visually engaging than, for example, microscope images or big-data plots. We have added the Tfh label to Figure 3C and are grateful that the Reviewer noticed the omission.

-It would be interesting for the authors to comment in the body of the text about the frequency of the risk variant. The MAF is between 5 and 10% (according to dbSNP) suggesting that the risk variant is present at 90 to 95% in the population. How do they explain such a great effect size in healthy controls for a variant that is so common? Which additional mechanism could actually lead to the disease?

RESPONSE: We thank the Reviewer for this point. Indeed, per the 1000 Genomes Project (GRCh37) the prevalence of the A risk allele is 95% in the European population, 85% in the East Asian population, and nearly 100% in the African population, with the C protective allele making up the remainder. As with most common variants, the effect size is relatively small (e.g. a decrease in Tph frequency from 3.9% to 2.7% of CD4+ T cells in healthy donors, Fig. 3D). Correspondingly, the effect on risk for incident RA is modest (odds ratio 0.74 for C vs. A allele). This information is included in the revised manuscript, including allele frequencies in multiple populations (**new Supplementary Table 1**). We postulate that a relative reduction in pathogenic Tph cells could contribute to this effect, though we cannot exclude the possibility that effects of this variant beyond those we have discovered may also contribute, as noted by Reviewer 1 and emphasized in the revised manuscript.

We highlight however that the goal of the work is not to explain the effect of a relatively uncommon protective variant but instead to use this variation as an "experiment of nature" to uncover a new mechanism in immunology – in this case, the previously unknown role of ICOS ligation in Tph development.

-Is there the same correlation between rs117701653 genotype and the different observed phenotypes (ICOS expression, Tph abundance...) in RA patients compared to healthy donors? And what about T1D?

RESPONSE: We appreciate this suggestion. The relatively low frequency of the protective variant renders targeted recruitment of RA and T1D patients infeasible, even in the >50,000-donor Mass General Brigham Biobank, and especially given potential confounding by disease activity and treatment. However, we do not regard this as a limitation of the work, because – as noted above – the manuscript seeks to define a variant associated with elevated risk for RA and to use normal variation to discover novel immunobiology.

Reviewer #3 (Remarks to the Author):

This is a very nice study which biochemically delineated the impact of rs117701653. Authors show preferred genotype dependent binding of SMCHD1, and ensuing levels ICOS and Tph frequency. I think this work nicely documents the elusive role for non-coding variants.

RESPONSE: We are grateful for these very positive comments.

Major comments:

1) I think clarity of nomenclature should be addressed. I understand that it can be lower or enhanced risk, but I think title and language should be clear in calling the allele studied as a protective allele first and foremost.

RESPONSE: we thank the Reviewer for this suggestion. In the revised manuscript, we make clear throughout that the minor allele C is protective and the major allele A confers risk. We note however that the most interesting finding in the manuscript concerns the risk allele A, which we show drives higher ICOS expression that in turn promotes Tph development. Our focus therefore remains on the risk allele, not the protective allele.

2) Also genetically it should be clearly spelled out that the biggest effect is seen in homozygosity of CC as a protective allele. Homozygosity in all populations per Gnomad is actually very rare, compared to what one could expect from rate of heterozygosity, which suggests some form of purifying selection. Thus there may have been a price to be paid for this protection (and should be discussed in light of rare genetics known.

RESPONSE: We thank the Reviewer for the intriguing suggestion that the rs117701653 C allele could be undergoing purifying selection. Recent studies have highlighted that low-frequency variants ($0.5\% < \text{MAF} < 5\%$), especially those regulating host-pathogen interactions, might be affected by evolutionary purifying selection.^{3,4} For example, rare homozygotes for TYK2 P1104A (rs34536443, a coding variant) impair the immune response to IL-23, which is critical for defending against tuberculosis infection, resulting in strong negative selection on homozygotes beginning 2,000 years ago.⁵ Moreover, CTLA4-ICOS locus has been under selection during Black Death caused by the *Yersinia pestis*, suggesting that rare homozygotes displaying low ICOS and T helper cell polarization after infection might have been experienced negative selection throughout evolutionary history, consistent with known roles of this locus in pathogen defense.^{6,7}

We tested whether the C allele or the proportion of C/C homozygotes was statistically unexpected given the allele frequencies and the assumption of Hardy-Weinberg Equilibrium (HWE). We performed a traditional HWE test for the GnomAD v3.1.2 dataset, but observed no significant deviation in the observed genotypes given the allele frequencies. These data do not confirm or reject the possibility of selection in the past, however. We have added this table as **new Supplementary Table 1** and added a related comment in the Discussion.

3) Figure 1 D. Positive control variance supersedes the actual (despite reasonable and statistically tight) results of rs117701653, which make me wonder if the finding is robust enough. Perhaps it should be confirmed in PHA blasts from rs117701653 carrying individuals.

RESPONSE: We thank the Reviewer for this comment. We recognize that there is more “noise” in the control A/C dataset (using the positive control HS17 promotor, where SMCHD1 binding is expected and thus binding in that group is higher than for rs117701653). We assessed whether the variances of the two groups were actually different using the binding fold enrichment normalized by the mean value of each group. We did not observe any difference (see **new Figure R1** below, for Reviser consideration only), supporting the likelihood that the “noise” was simply stochastic variation. While PHA effectively activates and proliferates primary T cells, we were concerned that such strong stimulation would mask the rs117701653 / SMCHD1 interaction effect due to the time-sensitive, stimulus-dependent regulation observed in primary T cells, as illustrated in Figures 5 and 6. However, we highlight that Figure 1D remains one of many convergent lines of evidence, including pulldown, supershift (3 antibodies), and *SMCHD1* gene targeting, as well as fully independent studies documenting a new role for ICOS in Tph development.

Figure R1. CHIP-qPCR targeting rs117701653 and positive control HS17 promotor

4) I think the correlations shown in figure 4 are nice, but I don't understand where the connection to rs117701653 is? Do patients with RA or T1D with rs117701653 have more or less Tph cells and all the ensuing downstream effects? Without this it very hard to interpret these results in the context of RA or T1D which was the point of this paper. At this point this rs117701653 is merely correlated and appears to cause with ICOS expression in healthy subjects.

RESPONSE: We regret that the importance of Figure 4 was not more evident and have revised the text to improve clarity. Figures 1 and 2 had established that allelic variation at rs117701653 modulates binding of SMCHD1 and that the rs117701653 / SMCHD1 interaction regulates expression of ICOS. In Figure 3, we identify correlations between allelic variation at rs117701653 and Tph abundance, as well as

between ICOS expression by memory CD4+ T cells and Tph abundance. Figure 4 confirms these relationships in RA patients, both in blood and in inflamed synovial fluid. These findings motivate Figures 5 and 6, which establish that ICOS regulates Tph development. Given the low abundance of the C allele, we are unable to explore genotype effects in RA (or in T1D), but we hope the analytical flow will clarify why this is not a limitation, since the healthy donor samples were sufficient to enable identification of the new ICOS-Tph connection.

Reviewer #4 (Remarks to the Author):

Kim et al. showed that the protective C allele of rs117701653 is involved in the repression of ICOS expression via binding of SMCHD1. Furthermore, they try to show that rs117701653 is involved in Tph cell differentiation through ICOS. Unfortunately, however, the experiments on differentiation have had procedural and experimental design problems and are insufficient to show its involvement in Tph differentiation. Furthermore, the involvement of SMCHD1 in Tph cell differentiation by rs117701653 has not been addressed. To reach the level of publication, it is required to address the following issues.

RESPONSE: We thank the Reviewer for careful reading of our manuscript and are attentive to suggestions for improvement.

#The definitions of positive and negative expression (such as FMO or isotype control) of molecules including ICOS and PD-1 in flow cytometry needs to be specified in gating strategy. If Isotype control antibodies are used in the definition, please add them to the table. Furthermore, a clear definition of PD-1(high), which is crucial for Tph and Tfh cells, needs to be described in this manuscript to ensure the reliability of the Tph and Tfh cell frequency data.

RESPONSE: We employed isotype controls to determine the background signal for CXCR5 and PD-1 expression. We considered cells with a higher signal level than the negative controls as positive for CXCR5 and PD-1 expression, and cells with a similar signal to the control as negative.

To distinguish cells with high and intermediate PD-1 expression among the positive cells in resting T cells, we established a threshold of 4,100 on a biexponential scale. Our gating strategy revealed that the frequency of PD-1-high population exhibited greater variability across individuals and genotypes compared to the frequency of PD-1-intermediate population (Figure 3C). For the differentiated Tph cells, we used a threshold of 9,700 on a biexponential scale to clearly distinguish cells with the PD-1-high expression after 3 days of differentiation (Figure 5A).

We have included the list of isotype controls in supplementary table and described the detailed gating strategy for the identification of CXCR5-PD-1-high and CXCR5+ PD-1-high cells in Methods.

#Fig5 is intended to show that ICOS signaling is important for Tph cell differentiation in vitro. However, the Tph cell differentiation culture conditions in Fig 5 and 6 may not be appropriate. Previous studies have shown that the Tph cell phenotype, PD-1 expression and CXCL13 production, can be induced under TGFβ alone conditions, but this is not reproduced by the conditions in this paper. (Fig 5B, C, and F).

Although it is known that about 20% of synovial CD4-positive cells in actual rheumatoid arthritis patients produce CXCL13, the MFI of CXCL13 and IL21 in Fig 5 and S8 is quite low (~300) compared to PD-1 (~100000) and other factors. Please show the positivity rate and a representative dot plot of CXCL13 and IL21 in FigS8.

Previous studies have often used plate bound CD3 antibodies and naive human CD4 T cells for Tph cell differentiation. Although the use of memory CD4 in Fig. 5 to investigate the role of ICOS signaling in Tph cell differentiation is reasonable, CD3 stimulation method needs to be optimized to achieve adequate Tph differentiation.

RESPONSE: We are grateful for this opportunity to describe the process by which we arrived at our method to differentiate Tph cells.

In **new Supplementary Figs. 10A-C and 10E**, we demonstrate that, as expected, culture in TGF- β alone does yield CXCR5-neg PD-1hi cells that produce IL-21 and CXCL13. We added the plots showing the frequency of IL-21 and CXCL13 in **new Supplementary Fig. 10 D,E,F**.

The Yoshitomi group reported that CXCR5- PD-1hi CD4+ T cells can be induced from naive CD4 T cells upon stimulation by plate-bound anti-CD3, soluble anti-CD28, TGF- β and neutralizing anti-IL-2.^{8,9} We tried these culture conditions using memory T cells, but found that the addition of anti-IL-2 antibody inhibited T cell proliferation. To optimize CXCR5- PD-1 high cells differentiation from memory T cells, we assessed several T cell stimulation methods for CXCR5- PD-1hi differentiation, including plate-bound anti-CD3 and soluble anti-CD28, ImmunoCult (StemCell), and Dynabead (Invitrogen) with TGF- β , without anti-IL-2 antibody. Plate-bound CD3 and soluble CD28 antibodies, as expected, induced CXCR5- PD-1hi cells that produced CXCL13 after 6 days of differentiation. However, after day 6, the restimulated cells did not proliferate and differentiate properly. On the other hand, ImmunoCult and Dynabead generated CXCR5- PD-1hi cells expressing IL-21 and CXCL13 during repeated stimulation. Notably, differentiated cells by Dynabead produced higher IL-21 and CXCL13 than those by ImmunoCult (Please see the figure below). Based on our optimization, we concluded that Dynabead stimulation is the most suitable method for in vitro Tph differentiation. We include these data here for Reviewer consideration only (**new Figure R2**).

Figure R2. Techniques of Tph differentiation (please see text for methods).

#Fig S8: PD-1 expression and CXCR5-PD-1(high) frequency are higher with ICOS stimulation on Day 3 and Day 18, but without ICOS stimulation on Day 6. It is expected that the improvement of induction conditions will lead to more consistent results throughout the study period. Is it reasonable to argue based solely on the results of Day 3 and Day 18?

RESPONSE: The Reviewer correctly notes that the effect of ICOS ligation on the appearance of the Tph phenotype *in vitro* is more evident during certain timepoints, a finding echoed in experimental replicates. We report the data as we observed them. We have no explanation for this timecourse, but consider the result concordant with multiple other lines of evidence that together define a rs117701653-SMCHD1-ICOS-Tph axis, including *in vivo* data from healthy and RA human donors.

#Fig6: As shown in Fig. 3 for, the frequency of ICOS-positive cells and Tph cells in memory CD4 T cells differ by genotype from the beginning day 0. The Day 3 results only reflect the Day 0 state in short differentiated cultures (Fig. 6A,B) and do not indicate that genotype is involved in the *in vitro* differentiation of Tph cells. Rather, it is better to culture naive CD4, which initially expresses little ICOS independent of the genotype (Fig 3B), in the *in vitro* differentiation. By differentiating cells with no difference in initial ICOS expression, it is possible to show that differences in ICOS levels caused by genotyping can affect Tph cell differentiation.

RESPONSE: Indeed, ICOS expression in memory T cells was significantly increased from day 0 (0.46-3.9%) to day 3 (84-98%) (Fig. 3B and Fig. 6A), leading us to believe that this initial ICOS difference had little influence on the overall genotype effect on inducing ICOS expression during Tph differentiation. However, to address the Reviewer's question directly, we evaluated the effect of ICOS ligation on the differentiation of naive T cells into Tph cells (**new Supplementary Fig. 11**). In contrast to memory cells, naive cells differentiated much less into CXCR5- PD-1hi cells and did not display any significant TGF- β and ICOS effect on IL-21 production at day 3. We conclude that memory T cells are more effective for generating CXCR5- PD-1 hi Tph cells and thus more likely relevant to the genotype effect.

#The involvement of SMCHD1 in Tph cell differentiation by #rs117701653 has not been addressed. Tph cell differentiation using SMCHD1 knockout or lentivirus/siRNA knockdown CD4T cells can be used to demonstrate SMCHD1 involvement.

RESPONSE: We thank the Reviewer for this suggestion. We applied CRISPR-gRNA to delete SMCHD1 in memory T cells derived from A/A and C/C individuals. The SMCHD1-deleted cells were differentiated for 3 days. C/C donors exhibited a significant increase in ICOS expression and IL-21 production. These findings support our identification of a rs117701653-SMCHD1-ICOS pathway that regulates Tph differentiation (**new Figs. 6G,H**).

#There are some discrepancies between the results of Fig2B and Fig2C. For example, in CD4+T, the expression of ICOS in AC is intermediate between in AA and CC, but in Jurkat qPCR, its expression in AC is as decreased as in CC. Furthermore, while there is no difference in CD28 mRNA in CD4, protein level of CD28 in Jurkat shows a decrease in CD28 in CC and p values are lacking. Discussion about these discrepancies is required.

RESPONSE: We thank the Reviewer for these comments. Certainly multiple factors distinguish primary polygenic CD4 T cells and Jurkat cells. We have added the p value for CD28 in Figure 2C as requested (protein level; A/A vs. A/C p=0.12, A/A vs. C/C p= 0.061) and highlighted these differences in Results, adding additionally a new functional analysis of signaling via CD28 vs. ICOS in response to the suggestion from Reviewer 1 (**new Supplementary Fig. 3**).

#One important issue of this paper is that ICOS stimulation is more involved in Tph cell differentiation than in Tfh cells, but the reasons for this are not discussed. For example, in Fig. 6S, about half of the Tfh cells from healthy subjects are negative for ICOS. Could this explain why ICOS stimulation differentiates Tph cells preferentially in human?

RESPONSE: We appreciate this excellent point. To accurately assess the ability of ICOS to drive precursors toward Tph or Tfh, an in vitro differentiation assay would be required using precursor cells. In our study, while Tph cells were constantly induced over time, the culture condition transiently induced Tfh cells at day3. This suggests that the conditions may not be the optimal for generating stable Tfh cells. Considering this, our focus was primarily on investigating the impact of ICOS on Tph rather than Tfh. However, we acknowledge the possibility of ICOS exerting an effect on Tfh cells. We have added a discussion of this possibility to the discussion section.

#Please show the ICOS positivity rate in each fraction including Tph cells and Tfh in RA as well as Fig6.

RESPONSE: We have added the frequency of ICOS+ Tph and Tfh in RA patients, now moved to **new Supplementary Figure 9**.

We thank the Reviewers and the Editors for their attention to our manuscript and hope that the revised manuscript has now addressed all concerns and is found worthy for *Nature Communications*.

Peter A. Nigrovic, MD, for the authors

1. Jansz, N. *et al.* Smchd1 regulates long-range chromatin interactions on the inactive X chromosome and at Hox clusters. *Nat Struct Mol Biol* **25**, 766-777 (2018).
2. Dean, A. In the loop: long range chromatin interactions and gene regulation. *Brief Funct Genomics* **10**, 3-10 (2011).
3. Gazal, S. *et al.* Functional architecture of low-frequency variants highlights strength of negative selection across coding and non-coding annotations. *Nat Genet* **50**, 1600-1607 (2018).
4. Kerner, G. *et al.* Genetic adaptation to pathogens and increased risk of inflammatory disorders in post-Neolithic Europe. *Cell Genom* **3**, 100248 (2023).
5. Boisson-Dupuis, S. *et al.* Tuberculosis and impaired IL-23-dependent IFN-gamma immunity in humans homozygous for a common TYK2 missense variant. *Sci Immunol* **3**(2018).
6. Klunk, J. *et al.* Evolution of immune genes is associated with the Black Death. *Nature* **611**, 312-319 (2022).
7. Kopf, M. *et al.* Inducible costimulator protein (ICOS) controls T helper cell subset polarization after virus and parasite infection. *J Exp Med* **192**, 53-61 (2000).
8. Kobayashi, S. *et al.* TGF-beta induces the differentiation of human CXCL13-producing CD4(+) T cells. *Eur J Immunol* **46**, 360-71 (2016).
9. Yoshitomi, H. *et al.* Human Sox4 facilitates the development of CXCL13-producing helper T cells in inflammatory environments. *Nat Commun* **9**, 3762 (2018).

REVIEWERS' COMMENTS

Reviewer #1 (Remarks to the Author):

Many thanks to the authors for addressing all of my comments. I have no further major concerns. However I would suggest they address in the discussion the following minor point. The new data shown in Fig. 6G and 6H show smaller differences between genotypes than Fig. 3B or the edited Jurkat cells. In the same experiment SMCHD1 knockdown in CC cells caused what looks like a 0.5x increase in expression of ICOS vs a 10x increase achieved in the Jurkat model. While the new experiments overall support the model proposed, they also point to other potential factors that might be at play in mediating the effect of the variation in the human cells.

Reviewer #2 (Remarks to the Author):

The authors have successfully addressed the comments and integrated them in the body of the text and supplementary data.

Reviewer #3 (Remarks to the Author):

I would like to thank to authors for clarifying some raised points.

I am a bit disappointed that "Given the low abundance of the C allele, we are unable to explore genotype effects in RA (or in T1D)". I think this is key to true clinical effect of your studies.

I also wish authors tried the suggested PHA blast experiment from rs117701653 carrying individuals, as maybe it would have strengthened their findings.

Reviewer #4 (Remarks to the Author):

The authors have addressed all the issues raised by the reviewers.

REVIEWERS' COMMENTS

Reviewer #1 (Remarks to the Author):

Many thanks to the authors for addressing all of my comments. I have no further major concerns. However I would suggest they address in the discussion the following minor point. The new data shown in Fig. 6G and 6H show smaller differences between genotypes than Fig. 3B or the edited Jurkat cells. In the same experiment SMCHD1 knockdown in CC cells caused what looks like a 0.5x increase in expression of ICOS vs a 10x increase achieved in the Jurkat model. While the new experiments overall support the model proposed, they also point to other potential factors that might be at play in mediating the effect of the variation in the human cells.

RESPONSE: We agree with the Reviewer and have added a comment to the Discussion as recommended. Reviewer #2 (Remarks to the Author):

The authors have successfully addressed the comments and integrated them in the body of the text and supplementary data.

RESPONSE: We are grateful for the Reviewer's guidance in improving our manuscript.

Reviewer #3 (Remarks to the Author):

I would like to thank to authors for clarifying some raised points.

I am a bit disappointed that "Given the low abundance of the C allele, we are unable to explore genotype effects in RA (or in T1D)". I think this is key to true clinical effect of your studies.

I also wish authors tried the suggested PHA blast experiment from rs117701653 carrying individuals, as maybe it would have strengthened their findings.

RESPONSE: We thank the Reviewer for the careful attention to our manuscript. The goal of our study is to identify a pathway implicated in disease using a non-coding SNP as an "experiment of nature", not to examine the population impact of the SNP itself, although we highlight that the risk variant here is the *common* variant not the rare one. The frequency of the SNP is not relevant to the clinical significance of the work, which is to define a new pathway contributing to RA and T1D, and not only in patients with the risk SNP. Since GWAS "hits" are by definition variants that favor the transition of a healthy individual into one with disease, the proper place to study them is in healthy donors, not those who have already converted, where the function of the allele may well play a different role, or even none at all.

Reviewer #4 (Remarks to the Author):

The authors have addressed all the issues raised by the reviewers.

RESPONSE: We thank the Reviewer for taking time to consider our manuscript.